# A nanosystem targeting tissue inhibitor of metalloproteinase-1 for continuous spatiotemporal idiopathic pulmonary fibrosis therapy

Chuyu Li [1,4], Guihong Lu[2,4], Hanlin Chen[1], Chenguang Wang[1], Zhongjie Wang[1], Ruiqi Ming[1], Shujun Liu[1] & Lili Huang [1,3] ✉

Idiopathic pulmonary fibrosis (IPF), a progressive, life-threatening disease marked by excessive collagen deposition, severe tissue injury, and dysregulated oxidative stress, poses a major threat to human health. Despite clinical advances, current therapies have limited anti-fibrotic efficacy. Here we show a reactive oxygen species (ROS)-responsive nanosystem targeting tissue inhibitor of metalloproteinase-1 (TIMP-1) for spatiotemporally precise IPF treatment. Anti-TIMP-1 antibodies (aT) are conjugated to mesenchymal stem cell-derived exosomes (Mexo) via ROS-cleavable phenylboronic acid ester linkers (cl), yielding Mexo-cl-aT. Following intratracheal administration, cl linkers are selectively cleaved by elevated ROS in the IPF microenvironment, enabling ROS scavenging while releasing Mexo and aT to mediate tissue repair and collagen degradation, respectively. We demonstrate that a single dose of Mexo-cl-aT exerts robust therapeutic efficacy against IPF in a bleomycin-induced mouse model of advanced-stage fibrosis, thereby validating this nanosystem as a safe and efficient candidate for next-generation IPF therapies.

Idiopathic pulmonary fibrosis (IPF) is a chronic and progressive interstitial lung disease[1–3]. The progressive accumulation of extracellular matrix (ECM) leads to increased stiffness of injured tissues, seriously impairing pulmonary function and quality of life in patients[4,5]. Approximately 5 million people worldwide suffer from IPF, with a median survival of 3–5 years post-diagnosis[6]. Pirfenidone and nintedanib are the only two U.S. Food and Drug Administration (FDA)-approved antifibrotic drugs that can slow the progression of IPF by inhibiting fibroblast activation[7,8]. Despite promising advances in the treatment of early-stage patients, they have limited efficacy for advanced-stage patients with massive collagen accumulation within the lesions. Moreover, both medications have been reported to cause adverse side effects on other organs or tissues, such as significant

gastrointestinal symptoms and other complications[9]. Therefore, a new therapeutic strategy is urgently needed to halt or reverse IPF progression.

Matrix metalloproteinases (MMPs), a family of zinc-dependent endopeptidases, play a critical role in the normal turnover of the ECM[10]. Although MMP levels have been reported to facilitate ECM protein degradation during pulmonary fibrosis, there is still excessive ECM deposition in IPF lesions. Recently, researchers have come to realize the importance of maintaining the balance of ECM metabolism homeostasis in the treatment of IPF, which is dynamically regulated by the protease system of MMPs/tissue inhibitors of metalloproteinases (TIMPs) rather than by MMPs alone[11]. The increased TIMPs, particularly TIMP-1, bind to various MMPs and inhibits their activity[12,13]. Therefore,

[1]School of Medical Technology, Beijing Institute of Technology, Beijing, PR China. [2]Center for Child Care and Mental Health (CCCMH), Shenzhen Children's Hospital, Shenzhen, PR China. [3]Tangshan Research Institute, Beijing Institute of Technology, Tangshan, PR China. [4]These authors contributed equally: Chuyu Li, Guihong Lu. ✉e-mail: llhuang@bit.edu.cn

blocking TIMP-1 may promote ECM degradation via MMP catalysis, thus restoring the compromised alveolar space. Nonetheless, to the best of our knowledge, the use of TIMP-1 as a novel therapeutic target for IPF to regulate MMP activity for ECM degradation has rarely been reported. In addition, current views suggest that pulmonary redox imbalance, stimulated by oxidative stress, not only drives the pathogenic activation of fibroblasts but also induces apoptosis of epithelial and endothelial cells during the fibrotic process, leading to the destruction of lung architecture[14]. Therefore, scavenging excess reactive oxygen species (ROS) in the IPF microenvironment and regenerating the disrupted alveolar epithelium and endothelium are indispensable for the reconstruction of disrupted fibrotic alveoli and the restoration of pulmonary function.

Mesenchymal stem cell-derived exosomes (Mexo) serve as innate nano-carriers, adept at targeting inflammation/injury sites and augmenting drug retention at the lesion site while maintaining a superior biosafety profile[15,16]. Furthermore, Mexo exhibit a spectrum of therapeutic properties, including anti-inflammatory and anti-fibrotic activities, as well as facilitation of tissue repair and regeneration[17].

Here, we aimed to verify the role of TIMP-1 in the pathogenesis of IPF and develop a combination therapy that targets both ECM metabolism imbalance and pulmonary redox imbalance by exploiting their synergism to reverse the progression of IPF and ameliorate lung function. We engineered Mexo with TIMP-1 antibodies (aT) via a phenylboronic acid ester bond-based ROS-cleavable linker (cl) (Fig. 1a), which is subject to cleavage by ROS in fibrotic lesions. We rationalized that upon ROS exposure, the smart nanosystem Mexo-cl-aT would rapidly scavenge excess ROS and release aT and Mexo to accelerate the clearance of ECM and promote lung repair in pulmonary fibrosis, respectively (Fig. 1b)[18,19]. Therefore, the inhaled nanoformulation treatment promoted the clearance of excessive ECM and alveolar re-epithelialization in an advanced-stage pulmonary fibrosis mouse model.

## Results

### High expression of TIMP-1 in IPF patients and bleomycin-induced pulmonary fibrosis mouse models

To study the correlation between TIMP-1 expression and IPF progression, the TIMP-1-positive area in lung tissues from IPF patients was investigated using immunohistochemical (IHC) staining (Fig. 2a, Table S1). The results showed that TIMP-1 expression, which was mainly localized in the extracellular matrix (Supplementary Fig. S1), increased by 4.7-fold compared with that in the lung tissues of non-IPF controls (Fig. 2b). Furthermore, a positive correlation (r = 0.5831) was observed between TIMP-1 expression and fibrosis progression, as identified by the representative fibrosis marker collagen type I alpha 1 chain (COL1A1) (Fig. 2c). Similarly, the TIMP-1-positive area in lung tissues showed a 4.4-fold increase in the bleomycin-induced pulmonary fibrosis mouse model, and a positive correlation (r = 0.7383) was observed with COL1A1 (Fig. 2d–f). Furthermore, enzyme-linked immunosorbent assay (ELISA) analysis revealed that the level of TIMP-1 in the bronchoalveolar lavage fluid (BALF) of bleomycin-induced pulmonary fibrosis mice increased by 36.9-fold, and that in the lung tissues increased by 2-fold (Supplementary Fig. S2a, b). These

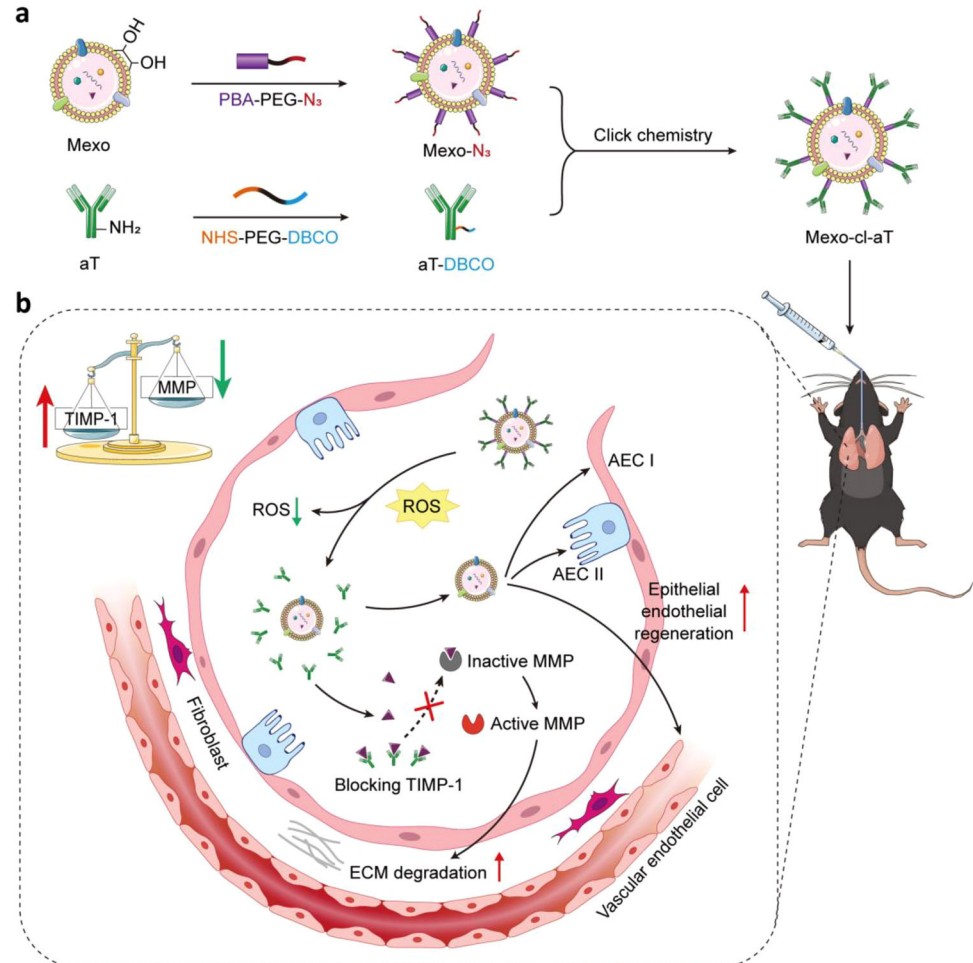

**Fig. 1 | Schematic illustration of the preparation and therapeutic mechanism of Mexo-cl-aT. a** Construction of Mexo-cl-aT. **b** Synergistic therapeutic mechanism of Mexo-cl-aT. (Created in BioRender. Zhang, F. (2025) https://BioRender.com/pinhfvr).

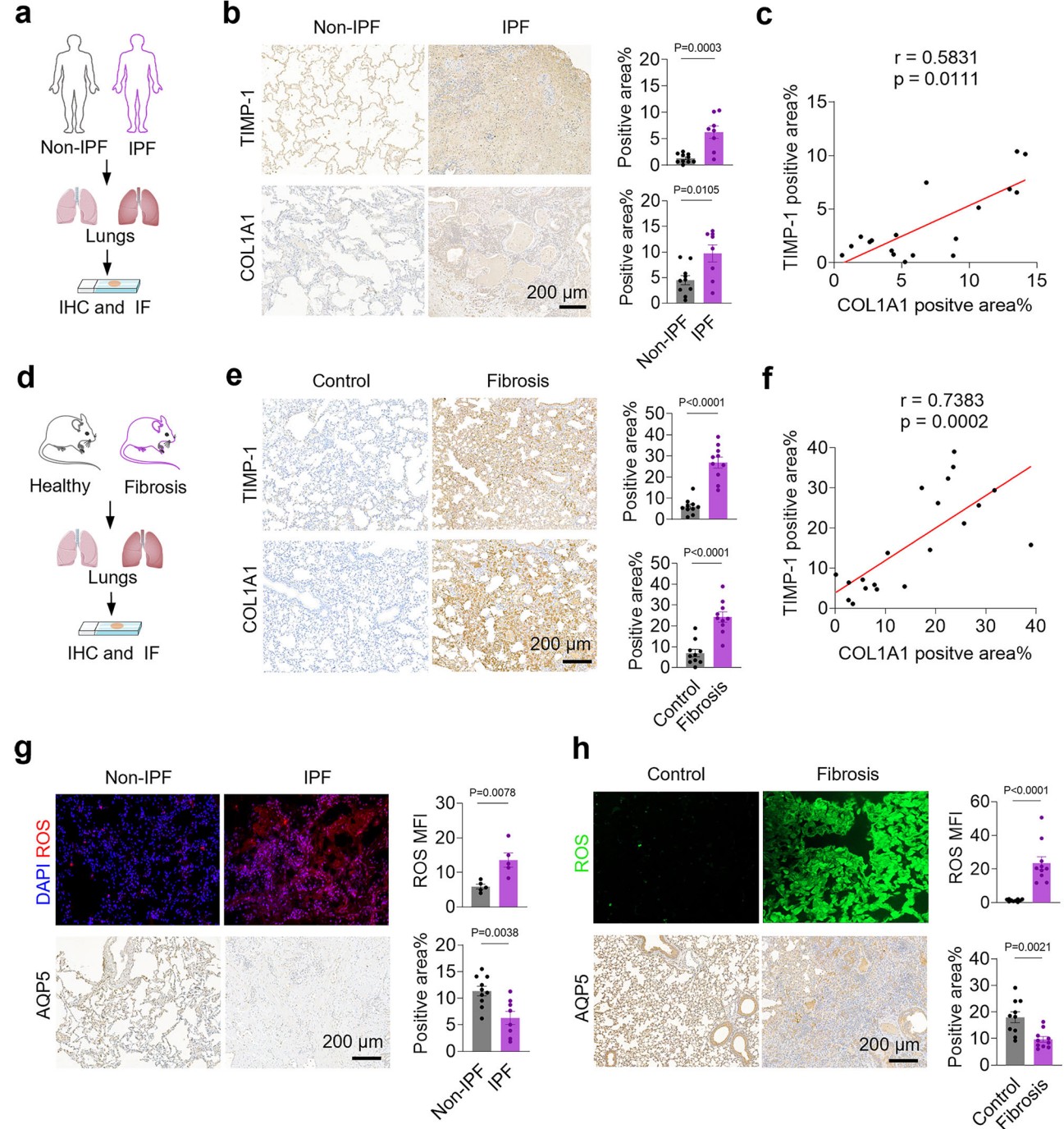

**Fig. 2 | Correlation of TIMP-1, COL1A1, ROS and AQP5. a** Schematic illustration of the experimental process. (Created in BioRender. Zhang, F. (2025) https://BioRender.com/24749x6). **b** IHC staining and quantification of TIMP-1 and COL1A1 in lung sections from non-IPF controls (n = 10 non-IPF patients) and IPF patients (n = 8 IPF patients). **c** Correlation analysis of TIMP-1 and COL1A1. **d** Schematic illustration of the experimental process. (Created in BioRender. Zhang, F. (2025) https://BioRender.com/chpxtxr). **e** IHC staining and quantification of TIMP-1 and COL1A1 in lung sections from controls and bleomycin-induced fibrosis mouse models. **f** Correlation analysis of TIMP-1 and COL1A1 (n = 20 samples). Immuno-fluorescence (IF) staining of ROS, IHC staining of AQP5, and the quantification of positive area in lung sections from non-IPF controls (n = 5 non-IPF patients for ROS; n = 10 non-IPF patients for AQP5) and IPF patients (n = 10 IPF patients for ROS; n = 8 IPF patients for AQP5) (**g**), as well as bleomycin-induced fibrosis mouse models (n = 10 mice) (**h**). Data indicate mean ± SEM, two-tailed unpaired Student's t test. Source data are provided as a Source Data file.

results reveal the prominent role of TIMP-1 in pulmonary fibrosis pathology, suggesting that TIMP-1 serves as a potential therapeutic target for pulmonary fibrosis.

Given that cellular oxidative stress and alveolar epithelial damage also contribute significantly to IPF pathogenesis, we assessed ROS levels and expression of the type I alveolar epithelial cell (AEC I) marker, aquaporin 5 (AQP5), in lung tissues[14,20]. As shown in Fig. 2g, h, the levels of ROS increased (2.3- and 16.4-fold, respectively) and AQP5 levels decreased significantly (1.8- and 1.9-fold, respectively) in the lung tissues of both IPF patients and mice. Moreover, we found strong correlations between TIMP-1, COL1A1, and AQP5 in mouse samples through quantitative analysis of IHC images, suggesting that TIMP-1, collagen deposition, and alveolar epithelial damage are synergistic in IPF progression (Supplementary Fig. S2c).

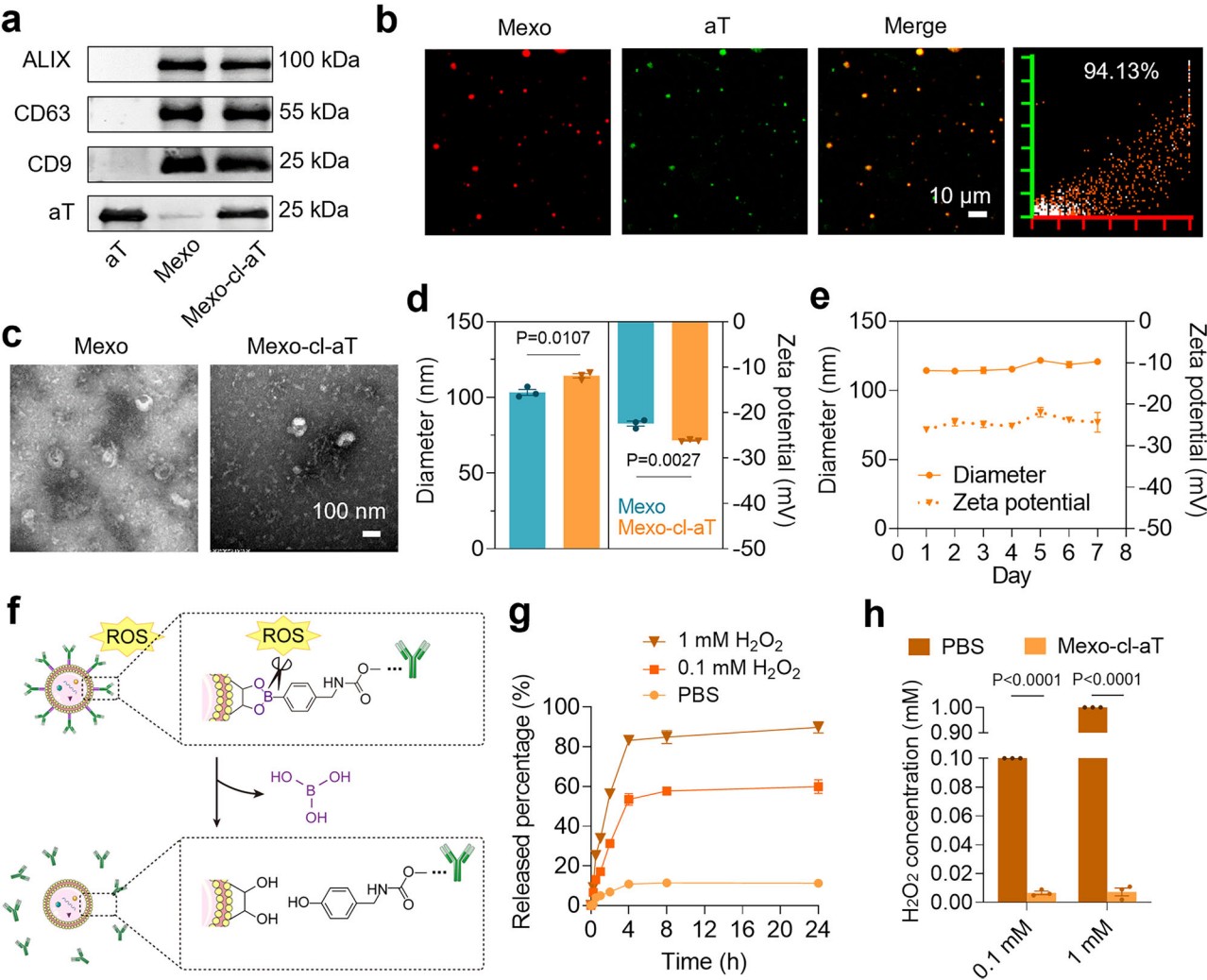

**Fig. 3 | Characterization of Mexo-cl-aT. a** Western blotting analysis of aT, Mexo and Mexo-cl-aT. **b** Confocal laser scanning microscope (CLSM) images and colocalization analysis of Mexo-cl-aT. Mexo were labeled with 1,1'-dioctadecyl-3,3,3',3'-tetramethylindocarbocyanine perchlorate (DiI), and aT were labeled with fluorescein isothiocyanate (FITC)-conjugated secondary antibody. **c** TEM images of Mexo and Mexo-cl-aT. **d** Diameter and Zeta potential of Mexo and Mexo-cl-aT. **e** Stability of Mexo-cl-aT under storage at 4 °C for 7 days. **f** Illustration of the ROS-

responsive release of aT from Mexo-cl-aT. (Created in BioRender. Zhang, F. (2025) https://BioRender.com/6uh3sgx). **g** Release curve of aT from Mexo-cl-aT treated with PBS, 0.1 mM $H_2O_2$ or 1 mM $H_2O_2$. **h** $H_2O_2$ concentration in solution without/with Mexo-cl-aT treatment. Data indicate mean ± SEM; two-tailed unpaired Student's t test. Measurements in Fig. 2a, b, d, e, g, h were taken from 3 biological replicates. Source data are provided as a Source Data file.

## Preparation and characterization of Mexo-cl-aT

The abovementioned results motivated us to develop a system to synergize ECM degradation, ROS scavenging, and tissue repair. For this purpose, we fabricated Mexo-cl-aT by conjugating aT with Mexo via an ROS-cleavable linker (cl). First, $N_3$-modified Mexo (Mexo-$N_3$) was obtained via the reaction between phenylboronic acid (PBA) and the glycosyl moieties on the Mexo surface proteins (Supplementary Fig. S3), which were subsequently conjugated with dibenzocyclooctyne (DBCO)-modified aT (aT-DBCO) (Supplementary Fig. S4) to yield Mexo-cl-aT[21,22]. Western blotting revealed that Mexo-cl-aT was composed of Mexo and aT, which were characterized by the presence of exosome biomarkers (ALIX, CD63, CD9) and the light chain of aT (Fig. 3a)[23,24]. The aT of Mexo-cl-aT were also verified by the binding of secondary antibody-linked Au nanoparticles (Supplementary Fig. S5). Moreover, the high colocalization (94.1%) of the aT and Mexo signals further confirmed successful construction of Mexo-cl-aT (Fig. 3b).

When the ratio of Mexo to aT was 5:1 (w/w), the Mexo-cl-aT displayed the greatest antifibrotic effect against transforming growth factor-β1 (TGF-β1)-induced primary mouse lung fibroblasts (MLFs) without causing cytotoxicity (Supplementary Fig. S6, S7), and the

loading capacity of aT was 89.9% at this time (Supplementary Fig. S8). Transmission electron microscope (TEM) images of Mexo-cl-aT showed no significant alteration in morphology compared to that of Mexo (Fig. 3c), indicating that aT modification did not influence the morphology of Mexo. After conjugation, the mean hydration particle size of Mexo-cl-aT slightly increased from 103.4 nm to 114.3 nm, while the zeta potential decreased (Fig. 3d). There were no detectable changes in particle size or zeta potential after storage in phosphate-buffered saline (PBS) at 4 °C for 7 days, suggesting the desired stability of the engineered Mexo-cl-aT for therapeutic applications (Fig. 3e).

In our design, the phenylboronic acid ester bond in Mexo-cl-aT is ROS-cleavable, leading to the release of free aT (Fig. 3f). To confirm this, $H_2O_2$ was exogenously added to the Mexo-cl-aT solution to mimic the high levels of $H_2O_2$ in the IPF pathological environment, and the released aT were quantified. As expected, the amount of free aT in the buffer solution gradually increased, reaching a plateau after 4 h of co-incubation, showing a strong time- and $H_2O_2$ concentration-dependent profile (Fig. 3g). Notably, more than 53.5% of the aT were released in the 0.1 mM $H_2O_2$ solution, and 83.2% were released in the 1 mM $H_2O_2$ solution, both significantly greater than the release

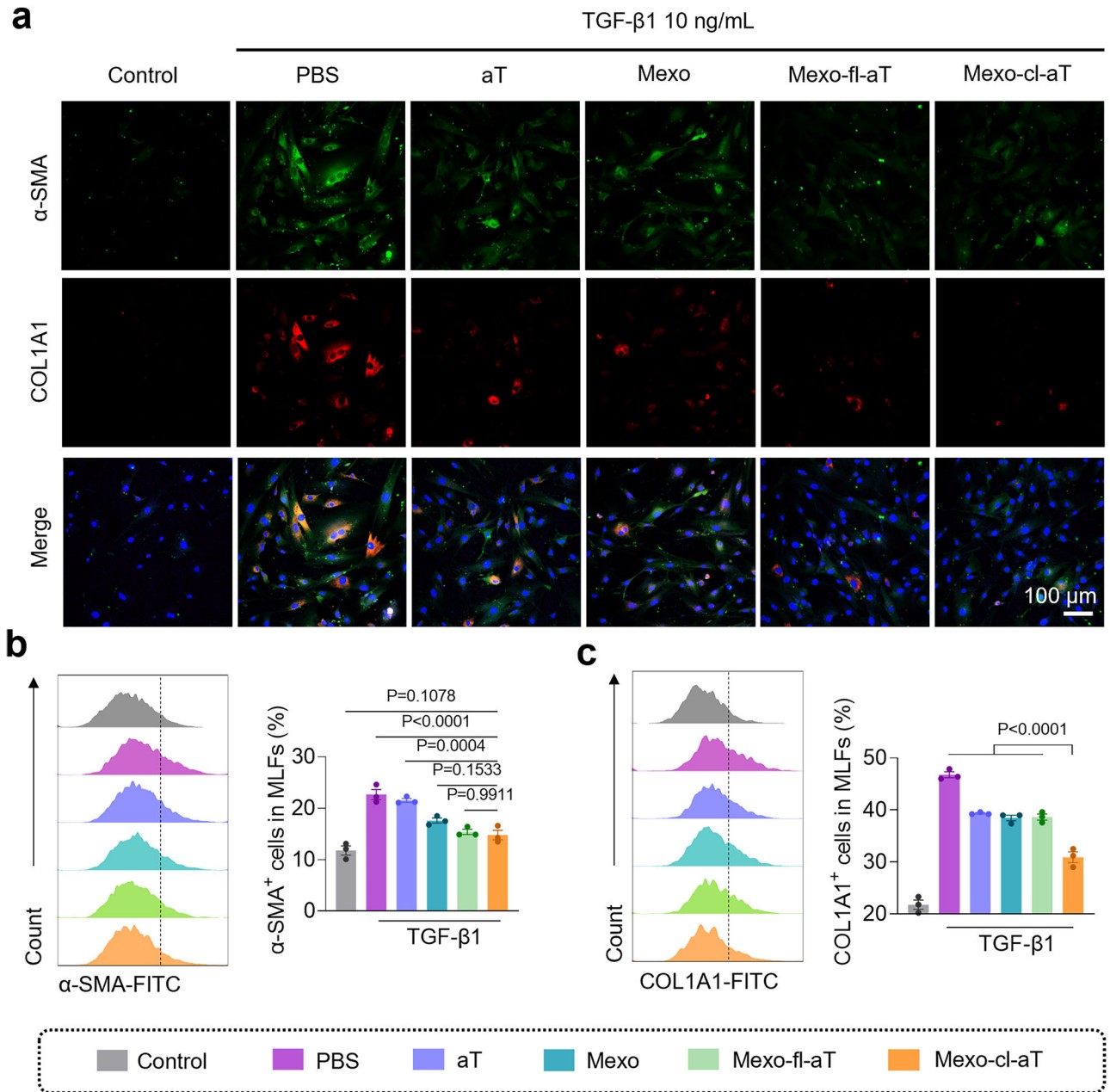

**Fig. 4 | Anti-fibrotic effects of Mexo-cl-aT in vitro. a** IF staining of α-SMA and COL1A1 in MLFs. Flow cytometry analysis and quantification of α-SMA⁺ (**b**) and COL1A1⁺ cells (**c**) in MLFs. Data indicate mean ± SEM; one-way ANOVA with Dunnett's multiple comparisons test; Measurements in Fig. 3a, b, c were taken from 3 biological replicates. Source data are provided as a Source Data file.

observed in PBS (10.8%). Moreover, $H_2O_2$ levels decreased by 90.0% after incubation with Mexo-cl-aT under both 0.1 mM and 1 mM $H_2O_2$ conditions (Fig. 3h). These results indicate that Mexo-cl-aT not only responds to ROS but also efficiently scavenges ROS.

## Anti-fibrotic effects and mechanisms of Mexo-cl-aT in vitro

Fibroblasts are recognized as one of the cellular sources of MMP-9, while activated fibroblasts are identified as the primary source of TIMP-1[12,25]. To explore the in vitro antifibrotic activity of Mexo-cl-aT, TGF-β1-induced primary MLFs were treated with different formulations for 24 h, and the myofibroblast marker alpha-smooth muscle actin (α-SMA) and COL1A1 were detected by IF staining[26]. As shown in Fig. 4a, the levels of α-SMA and COL1A1 expression were significantly upregulated in the PBS group, while aT or Mexo alone slightly inhibited α-SMA and COL1A1 expression. Better antifibrotic effects were observed when aT and Mexo cooperated. Notably, although Mexo-fl-aT (Mexo conjugated with aT via an ROS-uncleavable fixed linker) and Mexo-cl-aT did not differ in their ability to down-regulate α-SMA expression, the former was less effective than the latter in collagen degradation (Fig. 4b, c, Supplementary Fig. S9).

We further investigated the antifibrotic mechanisms of Mexo-cl-aT. Given that aT can block TIMP-1 and thus restore MMP activity (Fig. 5a), MMP-9 activity in the cell culture supernatant of each group was measured at 4 h post-treatment. As shown in Fig. 5b, c, both the aT and Mexo-cl-aT groups exhibited significantly decreased TIMP-1 levels and correspondingly increased active MMP-9 levels compared to the PBS group. In contrast, Mexo-fl-aT treatment failed to improve MMP-9 activity because the release of aT was prevented by the presence of the fixed linker. Similar results were also obtained in TIMP-1-silenced MLFs, confirming TIMP-1 as a key target of Mexo-cl-aT and demonstrating its

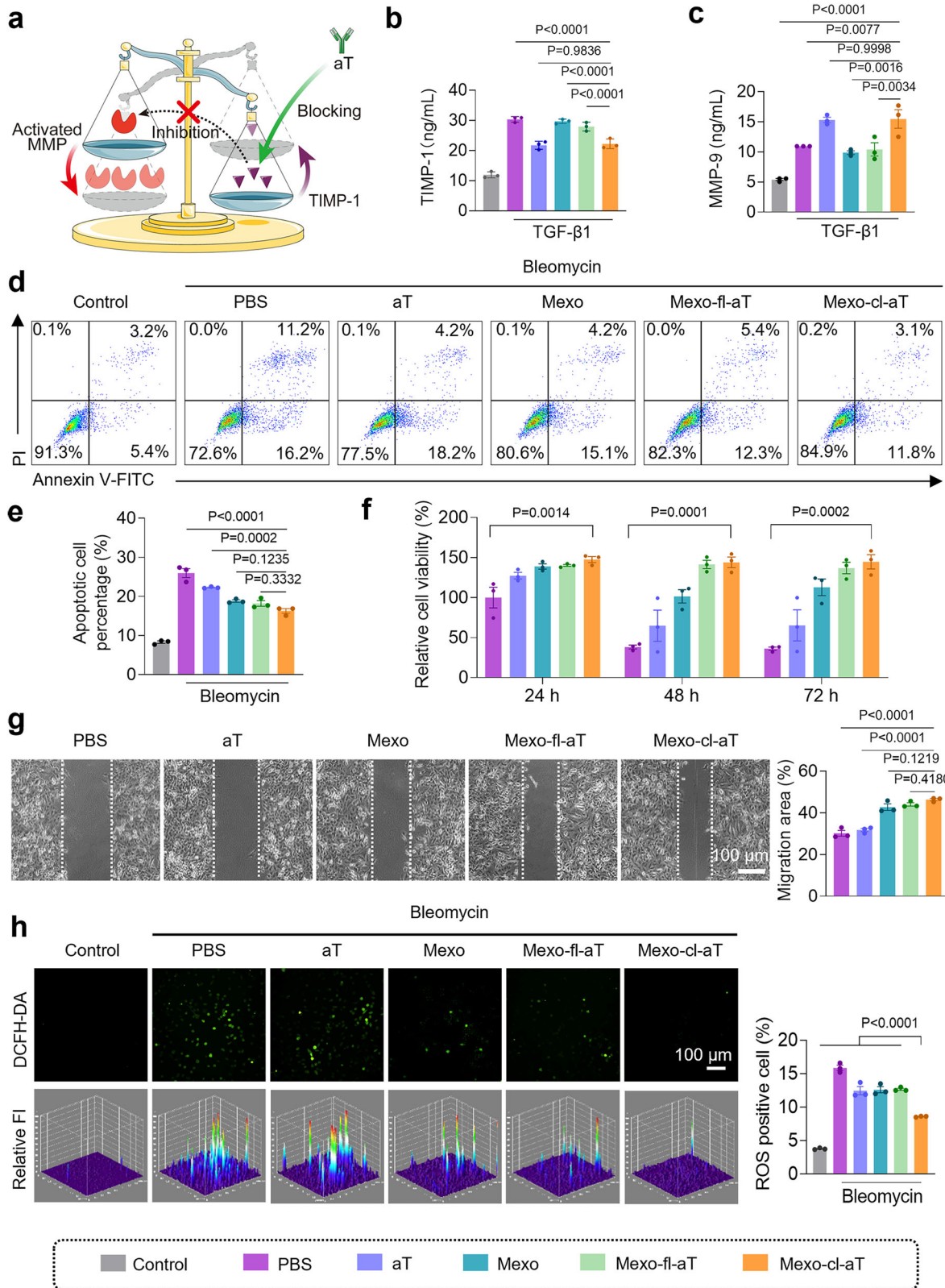

**Fig. 5 | The abilities of Mexo-cl-aT in MMP activation, ROS scavenging, alveolar epithelial repair in vitro. a** Illustration of the mechanism of aT for MMP/TIMP-1 rebalance. (Created in BioRender. Zhang, F. (2025) https://BioRender.com/5c51mwj). **b**, **c** The levels of TIMP-1 and active MMP-9 in MLF supernatants. **d**, **e** Cell apoptosis and quantification of A549 cells. **f** Cell viability of bleomycin-challenged A549 cells after various treatments for 24, 48, and 72 h. **g** Images of wound healing assay and quantification of migration area. **h** CLSM images, 3D maps and quantification of ROS in A549 cells. Data indicate mean ± SEM, one-way ANOVA with Dunnett's multiple comparisons test; Measurements in Fig. 4b–h were taken from 3 biological replicates. Source data are provided as a Source Data file.

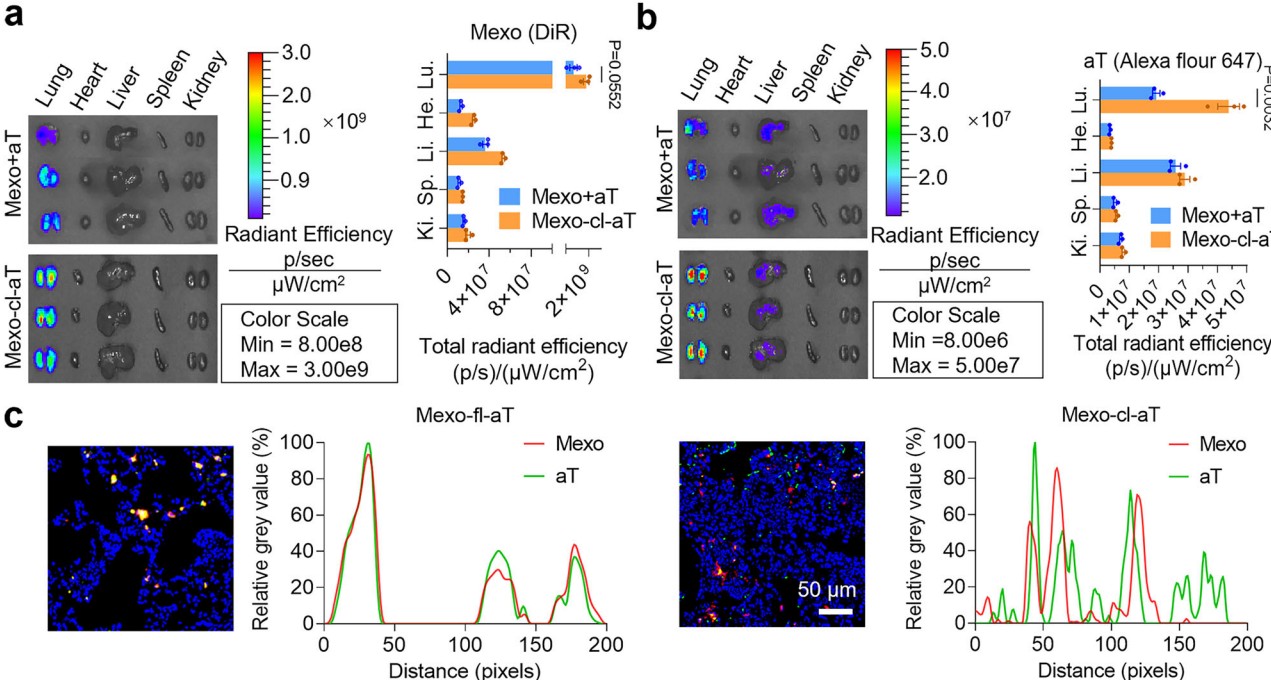

**Fig. 6 | Imaging of biodistribution and controlled release of Mexo-cl-aT.** Ex vivo imaging and quantitative analysis of Mexo (**a**) and aT (**b**) in the major organs harvested at 72 h after administration. The Mexo were labeled with DiR and the aT were labeled with Alexa Fluor 647-conjugated secondary antibody. **c** CLSM images and colocalization analysis of Mexo and aT in the cryosections of bleomycin-challenged mouse lungs. The Mexo were labeled with DiI and aT were labeled with FITC-conjugated secondary antibody. Data indicate mean ± SEM, measurements were taken from distinct samples; ns, not significant; two-tailed unpaired Student's t test. Measurements in Fig. 5a–c were taken from 3 biological replicates. Source data are provided as a Source Data file.

ability to shift the TIMP-1/MMP-9 balance toward an anti-fibrotic state (Supplementary Figs. S10–S12). It is important to note that while MMP-9 elevation may occur as an early, transient event in this reparative process, the sustained anti-fibrotic outcomes observed are primarily attributable to the rebalancing of the MMPs/TIMP-1 ratio achieved by aT, rather than prolonged MMP-9 activity alone.

Moreover, the percentage of apoptotic A549 cells in the Mexo-cl-aT group was reduced by 37.3% compared to the PBS group (Fig. 5d, e, Supplementary Fig. S13). Since the alveolar epithelium, which is primarily responsible for gas exchange, is vulnerable in IPF, we monitored the cell viability of alveolar epithelial A549 cells after treatment with different formulations daily for 72 h[27,28]. Cell Counting Kit-8 (CCK-8) assay showed that bleomycin induced clear time-dependent cytotoxicity to A549 cells, whereas Mexo-cl-aT treatment effectively maintained cell viability (Fig. 5f). Consequently, the Mexo-cl-aT group exhibited increased Ki67 expression, indicating that it promoted cell proliferation (Supplementary Fig. S14). To assess tissue repair capacity, we performed a wound healing assay where a scratch was used to mimic tissue injury. Representative images showed a 53.6% increase in the migration area with Mexo-cl-aT treatment versus PBS (Fig. 5g). Similar results were obtained for human umbilical vein endothelial cells (HUVECs) (Supplementary Fig. S15). Additionally, the ability of Mexo-cl-aT to scavenge ROS was assessed using CLSM and flow cytometry. As shown in Fig. 5h and Supplementary Fig. S16, compared to the PBS group, the Mexo, aT, and Mexo-fl-aT groups exhibited reduced ROS levels attributed to the anti-fibrotic effects of aT and/or the cellular repair capacity of Mexo. Notably, the Mexo-cl-aT group exhibited the weakest ROS signal, which decreased from 15.9% in the PBS group to 8.6%. This significant reduction suggests a substantial contribution of the phenylboronic acid ester bond to ROS scavenging, which was further confirmed by Amplex Red assay (Supplementary Fig. S17). These results collectively support the assertion that the therapeutic effects of Mexo-cl-aT are attributable to the synergistic effects of rapid clearance of excess ROS by the phenylboronic acid ester bond, MMP activation by aT, and cellular repair by Mexo.

### ROS-responsive behavior and metabolism of Mexo-cl-aT in vivo

The retention and extracellularly responsive release of aT in IPF lesions are critical for therapeutic efficacy. To this end, we separately labeled Mexo with 1,1-dioctadecyl-3,3,3,3-tetramethylindotricarbocyanine iodide (DiR), and aT with Alexa Fluor 647-conjugated antibodies. Subsequently, fluorescently labeled Mexo+aT (physical mixture) or Mexo-cl-aT was intratracheally injected into a bleomycin-induced pulmonary fibrosis mice, and their biodistribution was monitored using an in vivo imaging system (IVIS). Mexo signals peaked at 2 h and gradually declined by 72 h in both groups (Supplementary Fig. S18), indicating that more than half of the Mexo remained in the lungs for at least 72 h. At this point, the main organs were collected and imaged. The Mexo signals did not significantly differ between the two groups, indicating that the engineered modifications did not affect the normal metabolism of Mexo (Fig. 6a). In parallel, we found that the aT signals in the Mexo-cl-aT group were 2.3-fold stronger than those in the Mexo+aT group (Fig. 6b), indicating that the conjugation of aT with Mexo significantly improves the retention time of aT in the lungs.

To further verify the controllable release of aT from Mexo-cl-aT, we labeled Mexo with DiI and aT with FITC-conjugated secondary antibodies. After injecting fluorescently labeled Mexo-fl-aT or Mexo-cl-aT into bleomycin-induced pulmonary fibrosis mice, lung tissues were collected and imaged using CLSM. Since the fixed linker could not be cleaved by ROS in IPF lesions, aT were well colocalized with Mexo in the Mexo-fl-aT group. In contrast, the high level of $H_2O_2$ secreted in mouse lung tissues (Supplementary Fig. S19) led to the release of aT from Mexo in the Mexo-cl-aT group, which resulted in successful separation and poor colocalization (Fig. 6c). The results demonstrate that Mexo-cl-aT persists in IPF tissues and subsequently respond to ROS, releasing aT into the extracellular matrix of IPF lesions. This

validates our design strategy and lays the foundation for a highly efficient treatment of IPF.

### Therapeutic efficacy and safety of Mexo-cl-aT in vivo

Next, we evaluated the antifibrotic efficacy of Mexo-cl-aT in a bleomycin-induced pulmonary fibrosis mouse model (Fig. 7a). As shown in Fig. 7b, Mexo-cl-aT treatment made the greatest improvement in the macroscopic morphology of the lungs, as evidenced by the healthiest color and texture alongside minimal area of dark lesions. In contrast, all other treatment groups still exhibited varying degrees of impairment. Hematoxylin-eosin (H&E) staining, Masson staining, and Ashcroft score also revealed significant alveolar wall damage, tissue structural collapse, and collagen accumulation after bleomycin exposure. The antifibrotic effect was mild in Mexo or aT group, but it was enhanced when the two combined (Mexo+aT, Mexo-fl-aT and Mexo-cl-aT group). Notably, Mexo-cl-aT was more competent in restoring tissue structure and inhibiting ECM deposition, as evidenced by the recovery of the alveolar cavity and wall, and reduced collagen accumulation (Fig. 7c, d and Supplementary Fig. S20a), and the lowest levels of α-SMA and COL1A1 among all the treatment groups (Fig. 7e). Moreover, the average level of hydroxyproline (HYP, 280 μg/g of wet lung tissue) in the Mexo-cl-aT group approached that of healthy controls (286 μg/g of wet lung tissue) (Fig. 7f). These results strongly suggest the significant antifibrotic effects of Mexo-cl-aT in vivo.

In the dose optimization study, we further demonstrated that a single high-dose administration is particularly effective in restoring normal lung structure and function in bleomycin-challenged mice (Supplementary Figs. S21, S22). In a comparative study in which pirfenidone was used as a positive control, Mexo-cl-aT also exhibited pronounced anti-fibrotic efficacy (Supplementary Fig. S23), markedly attenuated leukocyte infiltration, and the decreased secretion of inflammatory cytokines such as IL-6, TNF-α, and IL-1β (Supplementary Fig. S24).

To assess the balance of ECM metabolism, we measured the activities of TIMP-1 and MMP-9 in BALF samples from healthy and fibrotic mice via ELISA. The results showed that TIMP-1 levels in mouse fibrotic lungs (PBS group) were significantly greater than those in healthy controls. Although free aT or Mexo reduced TIMP-1 levels and increased MMP-9 activity to some extent, this effect was enhanced when the two were combined. As expected, TIMP-1 levels in the Mexo-cl-aT group decreased by 90.3% compared to those in the PBS group, approaching levels seen in the healthy control group (Fig. 7g). Accordingly, Mexo-cl-aT treatment resulted in the greatest increase in MMP-9 activity among all treatment groups (Fig. 7h). Furthermore, the transcriptome analysis also revealed that Mexo-cl-aT treatment upregulated the expression of multiple MMPs and restored the balance of the MMPs/TIMPs protease system (Supplementary Fig. S25a).

To examine the ROS scavenging ability of Mexo-cl-aT in vivo, the ROS probes (DCFH-DA) were intratracheally injected into the lungs of bleomycin-challenged mice 4 h after the indicated treatments. Flow cytometry analysis revealed that fibrotic lungs had high levels of ROS, and Mexo-cl-aT treatment achieved an average ROS clearance of 76.8% (Fig. 7i and Supplementary Fig. S26). The transcriptomic analysis also revealed that Mexo-cl-aT treatment downregulated the expression of genes associated with oxidative stress, suggesting that effective alleviation of lung tissue damage (Supplementary Fig. S25b). Moreover, a marked decrease in TdT-mediated dUTP Nick-End Labeling (TUNEL) fluorescence was observed in the Mexo-cl-aT group (Fig. 7j). IHC images also showed increased expression of AQP5, prosurfactant protein C (ProSPC, a biomarker of type II alveolar epithelial cells), and von Willebrand factor (vWF, a biomarker of vascular endothelial cells) in lung sections after Mexo-cl-aT treatment (Supplementary Fig. S20b), suggesting its robust performance in tissue repair.

Finally, we assessed the biosafety of Mexo-cl-aT through hematologic and biochemical analysis, as well as H&E staining. Blood cell counts and hepatorenal functions were all within normal reference ranges (Supplementary Figs. S27, S28a). Additionally, the heart, liver, spleen, and kidneys of these mice showed no microscopic abnormalities in cell and tissue morphology (Supplementary Fig. S28b). All results indicate that Mexo-cl-aT does not cause any toxicity or apparent damage. In summary, our nanosystem exerts favorable synergistic therapeutic effects by rebalancing the protease system, scavenging ROS, and promoting tissue repair, while also providing good safety.

## Discussion

Currently, pirfenidone and nintedanib are used in the administration of patients with early IPF but are less effective and safe in patients with severe and advanced conditions[29]. One reason is that they have no direct effect on the excess collagen deposited in fibrotic lesions. TIMP-1 severely inhibits the activity of MMPs and significantly contributes to collagen deposition in IPF[12]. Therefore, aT remove the inhibitory effect of TIMP-1 on MMP activity and thus promote collagen degradation in fibrotic lesions through competitive binding to TIMP-1. Nevertheless, aT therapy carries the potential risk of over-activating MMPs, which relies on controlled drug delivery strategy to minimize side effects and improve bioavailability.

In our study, Mexo-cl-aT demonstrated efficient delivery of aT to the lesions in fibrotic lungs through four mechanisms. First, due to the slower elimination of exosomes compared with soluble proteins, conjugating aT to Mexo was found to prolong the half-life of aT by more than two-fold compared to aT alone. Second, Mexo, which have a natural ability to home to injury sites, are more inclined to linger in fibrotic lesions. Third, ROS responsiveness enables the extracellular release of aT, which facilitates its interaction with TIMP-1 in the ECM and thus efficiently regulates collagen metabolic homeostasis, resulting in an additional 20.4% reduction in lung collagen content compared to Mexo-fl-aT. Fourth, the drug release system regulates the concentration of free aT in response to ROS levels, which prevents MMP overactivation.

In addition to enabling targeted and responsive drug delivery, Mexo-cl-aT combines Mexo-mediated fibroblast inhibition, tissue repair, and the effects of aT to modulate antifibrotic responses. Due to the oxidative stress in the fibrosis microenvironment, the phenylboronic acid ester bond was chosen for its ability to not only be cleaved by ROS but also to scavenge them simultaneously. In summary, Mexo-cl-aT skillfully combines the powerful tissue repair capabilities of Mexo, the MMPs/TIMP-1 rebalancing of aT, and the ROS scavenging properties of the phenylboronic acid ester bond for continuous spatiotemporal IPF therapy, resulting in enhanced therapeutic effects.

Our nanosystem can be expanded for broader applications. Exosomes from different cell types could be explored for IPF therapy. For instance, exosomes derived from lung spheroid cells (LSCs) have been reported to be effective in treating IPF[30]. In cases of IPF caused by chronic viral infections, exosomes could be loaded with antivirals to facilitate pathogen clearance[31]. Beyond exosomes, future research could focus on TIMP-1 to further elucidate its various roles in ECM metabolism, cellular damage, inflammation, and repair. Some studies suggest that TIMP-1 may promote inflammation and cellular damage, indicating that targeting TIMP-1 could offer multiple benefits for IPF treatment[32–34]. Exosome-based aT therapies as accelerators of ECM degradation present a promising approach for treating various fibrotic diseases. In addition to fibrotic conditions, ECM metabolism also significantly impacts tumor genesis, progression, and treatment[13,35,36]. Therefore, developing therapeutic strategies that target TIMP-1 might enhance drug penetration and immune infiltration in tumor therapy.

We recognize several challenges that remain for the clinical application of Mexo-cl-aT. These challenges encompass the limited large-scale production and high variability of exosomes-persistent issues in bio-optimization, industrialization, and standardization.

 

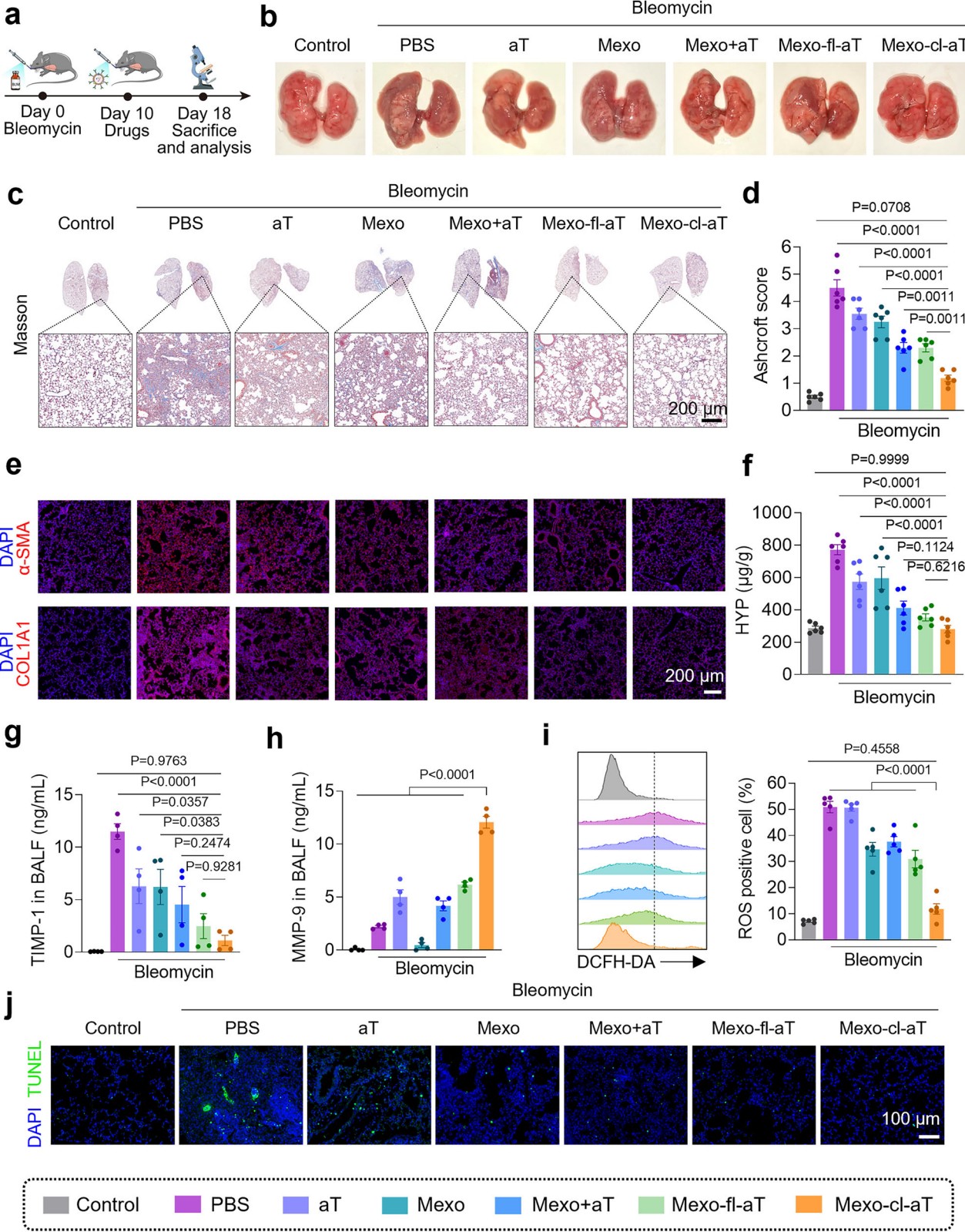

**Fig. 7 | Anti-fibrotic effects of Mexo-cl-aT in vivo. a** Illustration of animal experiment procedure. (Created in BioRender. Zhang, F. (2025) https://BioRender.com/15qup8o). Representative images (**b**), Masson staining (**c**), and Ashcroft score (**d**) of mouse lungs after different treatments (n = 6 per group). **e** IF staining of α-SMA and COL1A1 in mouse lung sections (n = 6 per group). **f** HYP content in mouse lung tissues (n = 6 per group). The level of TIMP-1 (**g**) and active MMP-9 (**h**) in mouse BALF (n = 4 per group). **i** Flow cytometry analysis of the ROS levels in mouse lung tissues (n = 5 per group). **j** TUNEL staining of mouse lung sections (n = 6 per group). Data indicate mean ± SEM; one-way ANOVA with Dunnett's multiple comparisons test. Source data are provided as a Source Data file.

Furthermore, the approach may not be optimized for all forms of fibrosis. For example, the antifibrotic potential of Mexo-cl-aT in a crystalline silica (CS)-induced fibrosis mouse model was limited, with large and fused fibrotic nodules still persisting in the lung tissues (Supplementary Fig. S29). This limitation is likely due to the difficulties in metabolizing CS particles in vivo, potentially leading to sustained and progressive pulmonary fibrosis.

## Methods

### Materials and agents

Triton X-100, bicinchoninic acid (BCA) protein assay kit, nanogold-conjugated secondary antibody, electrogenerated chemiluminescence (ECL), DCFH-DA, $H_2O_2$ assay kit, CCK-8, HYP content assay kit, mouse cytokine ELISA kits were purchased from Solarbio Science & Technology Co. Ltd. (Beijing, China). N-Hydroxy succinimide-(polyethylene glycol)$_4$-DBCO (NHS-PEG$_4$-DBCO), PBA-PEG$_4$-azide and NHS-PEG$_4$-azide were purchased from Xi'an Qiyue Biotechnology Co., Ltd. (Xi'an, China). DiI, DiR, and Amplex Red were purchased from Beyotime Biotechnology Co. Ltd. (Shanghai, China). Rabbit anti-mouse TIMP-1 antibody, Rabbit anti-mouse vWF antibody, Rabbit anti-human α-SMA antibody were purchased from ABclonal Technology Co. Ltd. (Wuhan, China). Rabbit anti-human CD63/CD9/ALIX antibody, HRP-conjugated goat anti-rabbit IgG(H+L) secondary antibody, Tetramethylrhodamine (TRITC)/FITC-conjugated goat anti-rabbit secondary antibody, rabbit anti-human TIMP-1 antibody, and mouse TIMP-1 ELISA kit were purchased from Proteintech (Wuhan, China). MMP-9 ELISA kit was purchased from Elabscience (E-EL-M3052, Wuhan, China). Bleomycin sulfate was purchased from Yuanye Bio-Technology Co. Ltd. (Shanghai, China). Silica was purchased from Shanghai Macklin Biochemical Co. Ltd. (Shanghai, China). Annexin V-FITC/PI apoptosis detection kit was purchased from Vazyme (Nanjing, China). Pirfenidone and recombinant mouse TGF-β1 was purchased from MedChemexpress (New Jersey, USA). Rabbit anti-mouse COL1A1 antibody, Alexa Fluor 488-conjugated rabbit anti-mouse α-SMA antibody and Alexa Fluor 647-conjugated goat anti-rabbit IgG secondary antibody, Rabbit anti-mouse AQP5 antibody were purchased from Thermo Fisher Scientific (Waltham, USA). Rabbit anti-mouse ProSPC antibody was purchased from Abcam (Cambridge, UK).

### Human specimens

Diseased lung tissue was harvested from IPF patients (diagnosed via standard criteria and MDT consensus) at the time of lung transplantation at the First Affiliated Hospital, Zhejiang University School of Medicine. Control lung samples (non-cancerous regions of lung cancer patients, or non-diseased areas of benign nodule patients) were collected for IHC and IF staining.

Patient demographics are summarized in Supplementary Table S1. All specimens were collected at the First Affiliated Hospital, Zhejiang University School of Medicine. Written informed consent was obtained from all donors prior to specimen collection. The study protocol was approved by the Clinical Research Ethics Committee of this hospital (Approval ID: 1047) and Shenzhen Children's Hospital (Approval ID: 2023159021).

### Cell culture

Human umbilical cord mesenchymal stem cells (hUMSCs) were obtained from the Cell Resource Center, Peking Union Medical College, and cultured in complete HMSC medium (SIMPSONLIFE, China). A549 cells and HUVECs were obtained from American Type Culture Collection (ATCC) and cultured in high glucose DMEM containing 10% FBS, penicillin (100 U/mL), and streptomycin (100 μg/mL) at 37 °C under 5% $CO_2$ environment. Primary MLFs were obtained from C57BL/6 J mouse lung tissues. Briefly, the tissues were harvested and cut into small pieces (1 mm³), digested in type IV collagenase (2 mg/mL) at 37 °C for 15–20 min, terminated and washed in high glucose DMEM

containing 20% FBS and antibiotics. Small pieces were spread on the petri dish (diameter 10 cm) and cultured in 2 mL complete medium for 24 h for stable adherence, followed by 4 mL medium replacement. Fibroblasts climbed out from the margin of small tissue on day 3. The fibroblasts were subcultured when the density reached 80%. Primary MLFs used for downstream experiments were between passages 3 and 5.

### Mexo isolation

hUMSCs were cultured in exosome-free medium. Culture supernatants were collected and centrifuged at $1000 \times g$ for 10 min and $2000 \times g$ for 20 min to remove cells and cell fragments, respectively. The obtained supernatant was centrifuged at $10,000 \times g$ for 60 min at 4 °C to remove debris and microvesicles. The final supernatant was then ultra-centrifuged at $100,000 \times g$ for 70 min to obtain exosome sedimentation. The obtained exosomes were resuspended in PBS. The protein concentration was determined by BCA.

### Preparation of Mexo-cl-aT and Mexo-fl-aT

Conjugation of NHS-PEG$_4$-DBCO to the aT. NHS reacted with the amino group of lysine on the antibody. NHS-PEG$_4$-DBCO was added to the aT at a molar ratio of 50: 1 (DBCO: aT). After incubation on shaker at RT for 1 h, the solution was filtered (Millipore, amicon ultra-0.5, 10 kDa) using centrifugation at $7000 \times g$ for 30 min to remove the excess NHS-PEG$_4$-DBCO. The purified DBCO-aT were suspended in PBS and analyzed using MALDI-TOF to confirm the purity and composition.

Conjugation of PBA-PEG$_4$-azide or NHS-PEG$_4$-azide to the Mexo. PBA and NHS reacted with saccharide groups and amino groups on the surface of the Mexo, respectively. Excessive PBA-PEG$_4$-azide was added to the Mexo at a concentration ratio of 100 μM: 1 mg/mL (PBA-PEG$_4$-azide: Mexo). After incubation at 4 °C for 2 h, the solution was filtered (Millipore, amicon ultra-0.5, 100 kDa) using centrifugation at $7000 \times g$ for 30 min to remove the excess PBA-PEG$_4$-azide. The purified Mexo-cl-azide was suspended in PBS. Mexo-fl-azide was prepared in the same way.

Conjugation of DBCO-aT to the Mexo-cl/fl-azide. DBCO-aT were conjugated with Mexo-cl/fl-azide by click chemistry reaction. After incubation of DBCO-aT and Mexo-cl/fl-azide at 4 °C for 2 h, the resulting Mexo-cl-aT or Mexo-fl-aT were washed and purified using ultracentrifugation, and suspended in PBS. Free DBCO-aT in the ultracentrifuged supernatant was determined using ELISA. Therefore, the capacity of aT on Mexo-cl-aT and Mexo-fl-aT could be calculated.

### Morphology of Mexo and Mexo-cl-aT

TEM images of Mexo and Mexo-cl-aT stained with antibody-linked gold nanoparticles (10 nm) were obtained using TEM (HITACHI, Japan) with an accelerating voltage of 80 kV. The size and zeta potential were determined using Zetasizer (Malvern, UK).

### Colocalization of Mexo-cl-aT

Mexo and aT were labeled with DiI and FITC-conjugated secondary antibody, respectively. Both dyes were added to the solution at a final concentration of 5 μM. Double-labeled Mexo-cl-aT were dropped on a glass slide, covered with a coverslip, and visualized using CLSM (Nikon, Japan). The fluorescence was also analyzed by flow nanoanalysis (Beckman, USA). Measurements were taken from distinct samples.

### Western blotting

All samples were denatured in boiling water bath for 10 min. Each sample containing 20 μg protein was loaded and separated by sodium dodecyl sulfate-polyacrylamide gel electrophoresis (SDS-PAGE). Exosomes were characterized using anti CD63, anti CD9, and anti-ALIX antibodies. The light chain of aT was characterized using HRP-conjugated secondary antibody. Target bands were observed by a gel imager (ThermoFisher, USA).

## Release assay in vitro

Mexo-cl-aT was labeled with FITC-conjugated secondary antibody. The labeled Mexo-cl-aT in 200 μL PBS containing 0 mM, 0.1 mM, or 1 mM $H_2O_2$, was added into dialysis bags (1000 kDa). Then, dialysis bags were immersed in 5 mL solution mentioned above, respectively. At predetermined time intervals, 200 μL liquid outside bags was removed for analysis and an equal volume of corresponding solution was supplemented. Finally, the absorbance at 488 nm of samples were measured using a microplate reader (BioTek, USA). Measurements were taken from distinct samples.

## $H_2O_2$ determination in vitro

The samples were collected at 4 h in the release assay. The content of $H_2O_2$ was determined using $H_2O_2$ assay kit according to the manufacturer's instructions. Briefly, $H_2O_2$ reacts with titanium sulfate to form a yellow titanium peroxide complex, which has characteristic absorption at 415 nm. Finally, the absorbance at 415 nm of samples was measured using a microplate reader. Measurements were taken from distinct samples.

## Determination of $H_2O_2$ in vitro (Amplex Red assay)

A549 cells were seeded on confocal dishes ($5 \times 10^5$ per dish) and incubated overnight. First, the cells were injured by bleomycin and then were co-incubated with various groups (PBS, aT, Mexo, Mexo-fl-aT and Mexo-cl-aT) for 48 h. Normal A549 cells were as the control group. Next, remove the culture medium and add 100 μM Amplex Red and 0.25 U/mL HRP. After 30 min, transfer the supernatant to a 96-well plate and read the absorbance at 570 nm using a fluorescence microplate reader.

## ROS detection in vitro

A549 cells were seeded on confocal dishes ($5 \times 10^5$ per dish) and incubated overnight. First, the cells were injured by bleomycin and then various groups (PBS, aT, Mexo, Mexo-fl-aT and Mexo-cl-aT) were co-incubated for 48 h. Normal A549 cells were as the control group. Next, the ROS probe DCFH-DA (10 μM) was added into the dishes and incubated with cells in FBS-free DMEM at 37 °C under 5% $CO_2$ in the dark for 20 min. Last, the level of ROS was imaged and quantified by CLSM and its analytical system. 3D maps of relative fluorescence intensity were produced by ImageJ software. For flow cytometry (BD FACSAriaII, USA), treated cells were digested from the dishes and incubated with DCFH-DA (10 μM) in the same conditions above. Measurements were taken from distinct samples.

## Anti-apoptosis effects in vitro

A549 cells were seeded on 6-well plates ($2 \times 10^6$ per well) and incubated overnight. First, the cells were injured by bleomycin and then various groups (PBS, aT, Mexo, Mexo-fl-aT and Mexo-cl-aT) were co-incubated for 48 h. Normal A549 cells were as the control group. Next, cells were digested from the dishes using EDTA-free trypsin, and Annexin V-FITC/PI staining was performed using apoptosis detection kit. Last, the cell apoptosis was analyzed by flow cytometry. Measurements were taken from distinct samples.

## Cell viability in vitro

The proliferation ability and viability of A549 cells and HUVECs treated with various groups (PBS, aT, Mexo, Mexo-fl-aT, Mexo-cl-aT) was evaluated by CCK-8. First, A549 cells and HUVECs were seeded in 96-well plates ($2.5 \times 10^3$ per well) and injured by bleomycin for 48 h. Next, medium with bleomycin was removed and replaced with fresh DMEM with 2% FBS. Then, various groups (PBS, aT, Mexo, Mexo-fl-aT, Mexo-cl-aT) were added. Last, CCK-8 assays were performed on 96-well plates cultured for 24, 48, and 72 h, respectively. The absorbance at 450 nm was detected using a microplate reader (Thermo Scientific, USA). Measurements were taken from distinct samples.

## Cell proliferation in vitro

A549 cells and HUVECs were seeded on confocal dishes ($5 \times 10^5$ per dish) and incubated overnight. First, the cells were injured by bleomycin for 48 h. Normal cells were as the control group. Next, bleomycin was removed and replaced with fresh DMEM containing 2% FBS, then various groups (PBS, aT, Mexo, Mexo-fl-aT and Mexo-cl-aT) were added and co-incubated for 72 h. Last, Ki67 staining was performed, and images were captured using CLSM. Measurements were taken from distinct samples.

## Wound healing assay in vitro

The proliferation and migration ability of A549 cells and HUVECs treated with various groups (PBS, aT, Mexo, Mexo-fl-aT, Mexo-cl-aT) was evaluated by wound healing tests. First, A549 cells and HUVECs were seeded in six-well plates ($2 \times 10^6$ per well) and incubated until the fusion of cells reached 100%. Next, a scratch line was created in the middle of each well using a 20 μL pipette tip, and then cells were washed three times with PBS to remove suspended cells. The cells were treated with various groups (PBS, aT, Mexo, Mexo-fl-aT, Mexo-cl-aT) and cultured in DMEM with 2% FBS for 24 h. Last, the cells were imaged by inverted fluorescence microscope (Olympus, Japan) to observe the area of the scratch. Measurements were taken from distinct samples.

## Determination of MMP-9 activity

For in vitro study, Primary MLFs were seeded on 12-well ($5 \times 10^5$ per well) and incubated overnight. First, the cells were activated by TGF-β1 (10 ng/mL) for 24 h. Next, various groups (PBS, aT, Mexo, Mexo-fl-aT and Mexo-cl-aT) were added and co-incubated with cells for 4 h. Then, the cultural supernatant was collected and added 1 mM PMSF to prevent protein degradation followed by centrifugation at $2000 \times g$ for 10 min at 4 °C to remove cells and cell debris. For in vivo study, mouse BALF samples were collected and centrifuged to remove cells and cell debris. Finally, MMP-9 activity was determined by ELISA. The absorbance at 450 nm was detected using a microplate reader. Measurements were taken from distinct samples.

## Anti-fibrotic effects in vitro

Primary MLFs were seeded on confocal dishes ($5 \times 10^5$ per dish) and incubated overnight. First, the cells were activated by TGF-β1 (10 ng/mL) and then various groups (PBS, aT, Mexo, Mexo-fl-aT and Mexo-cl-aT) were co-incubated for 24 h. Normal MLFs were as the control group. Next, cells were washed with PBS, fixed in 4% paraformaldehyde, permeabilized with 0.1% Triton X-100, and then blocked with 5% BSA before incubation in proper order with COL1A1 primary antibodies, TRITC-conjugated secondary antibodies, and Alexa Fluor 488-conjugated α-SMA antibodies. The cell nucleus was stained with Hoechst 33342. Last, the images were captured using CLSM. For the quantification of α-SMA$^+$ and COL1A1$^+$ cells in MLFs, treated cells were digested from the dishes, stained with the corresponding fluorescent antibodies, and analyzed by flow cytometry. Measurements were taken from distinct samples.

## Animals

C57BL/6J male mice (8–10 weeks) were purchased from Laboratory Animal Center, Peking University. Mice were anesthetized and received a single intratracheal injection of bleomycin sulfate at a dose of 2 units/kg. Mice challenged with bleomycin for 10 days were used for all subsequent animal experiments in vivo. For in vivo therapeutic studies, PBS, aT, Mexo, Mexo+aT, Mexo-fl-aT and Mexo-cl-aT (200 μg/mL Mexo, 40 μg/mL aT) in 50 μL solution were given one dose intratracheal injection in pulmonary fibrosis mouse lungs. Pirfenidone was as the positive control (p.o. 300 mg/kg, 100 μL, once daily from day 10–17). Mouse lungs were harvested and analyzed on day 18. For silica-induced pulmonary fibrosis model, mice challenged with silica for 2 weeks were used for all subsequent animal experiments in vivo. PBS,

aT, Mexo, Mexo+aT, Mexo-fl-aT and Mexo-cl-aT (200 μg/mL Mexo, 40 μg/mL aT) in 50 μL solution were given one dose intratracheal injection in pulmonary fibrosis mouse lungs. Mouse lungs were harvested and analyzed on week 4. All the procedures were approved by the Animal Ethics Association and the Ethics Committee of Beijing Institute of Technology (approval ID: 2022-0009-167).

### Determination of cytokines in mouse lung tissues

First, the tissues were washed with pre-cooled PBS, weighed, and chopped up. Next, chopped tissues were added to PBS (1 mL PBS for 0.1 g tissues), and PMSF was added to prevent protein degradation. The tissue samples were transferred to a homogenizer and ground thoroughly on ice. Finally, the homogenate was centrifuged at 2–8 °C, $5000 \times g$ for 10 min. The cytokines (TIMP-1, TGF-β, IL-6, IL-1β, TNF-α) in the supernatant were determined by ELISA.

### Transcriptomics

Total RNA was extracted from the mouse lung tissues using TRIzol® Reagent according the manufacturer's instructions (Magen). RNA samples were detected based on the A260/A280 absorbance ratio with a Nanodrop ND-2000 system (Thermo Scientific, USA), and the RIN of RNA was determined by an Agilent Bioanalyzer 4150 system (Agilent Technologies, CA, USA). Only qualified samples will be used for library construction. Paired-end libraries were prepared using a ABclonal mRNA-seq Lib Prep Kit (ABclonal, China) following the manufacturer's instructions. The mRNA was purified from 1 μg total RNA using oligo (dT) magnetic beads followed by fragmentation carried out using divalent cations at elevated temperatures in ABclonal First Strand Synthesis Reaction Buffer. Subsequently, first-strand cDNAs were synthesized with random hexamer primers and Reverse Transcriptase (RNase H) using mRNA fragments as templates, followed by second-strand cDNA synthesis using DNA polymerase I, RNAseH, buffer, and dNTPs. The synthesized double stranded cDNA fragments were then adapter-ligated for preparation of the paired-end library. Adaptor-ligated cDNA was used for PCR amplification. PCR products were purified (AMPure XP system) and library quality was assessed on an Agilent Bioanalyzer 4150 system. Finally, sequencing was performed with an Illumina Novaseq 6000/MGISEQ-T7 instrument. The data generated from Illumina/BGI platform were used for bioinformatics analysis. Differential expression analysis was performed using the DESeq2, DEGs with |log2FC| > 1 and Padj < 0.05 were considered to be significantly different expressed genes.

### $H_2O_2$ determination in vivo

Mice were intratracheally injected with 50 μL PBS or bleomycin sulfate (2 U/kg). After 10 days, the mouse lung tissues were collected and immediately homogenized in pre-cooled acetone (tissue: acetone = 0.1 g/mL). The process was operated on ice. The samples were centrifuged at 8000 rpm for 10 min at 4 °C, and the supernatants were obtained. The content of $H_2O_2$ was determined using $H_2O_2$ assay kit according to the manufacturer's instructions. Briefly, $H_2O_2$ reacts with titanium sulfate to form a yellow titanium peroxide complex, which has characteristic absorption at 415 nm. Finally, the absorbance at 415 nm of samples was measured using a microplate reader. Measurements were taken from distinct samples.

### Imaging of drug metabolism

For in vivo imaging, Mexo were labeled with DiR and aT were labeled with Alexa Fluor 647-conjugated secondary antibody. Bleomycin-induced pulmonary fibrosis mice were given intratracheal injection of Mexo+aT (simple mixed) and Mexo-cl-aT. Images were captured at time point 0, 2, 8, 24, 48, 72 h after injection by animal in vivo imaging system (PerkinElmer, USA). The major organs were dissected and subjected for ex vivo imaging. Average radiant efficiency of DiR and Alexa Fluor 647-conjugated secondary antibody was quantified by

Living Image analysis software. For confocal microscopy, Mexo were labeled with DiI and aT were labeled with FITC-conjugated secondary antibody. Bleomycin-induced pulmonary fibrosis mice were given intratracheal injection of Mexo-fl-aT and Mexo-cl-aT. After 4 h, the lungs of mice were collected, fixed in 4% paraformaldehyde for 24 h, dehydrated in 30% sucrose solution for 1–3 days and stored in the optimum cutting temperature (OCT) compound at −20 °C. The frozen lungs were further cut into 10 μm slices using freezing microtome (RWD Life Science, China). The sections were imaged using CLSM to analyze the controlled release of drugs.

### Histology analysis

Mouse lung tissues were collected, washed with PBS, and fixed in 4% paraformaldehyde for 24 to 48 h. For H&E, Masson's trichrome and picrosirius red staining, lungs were dehydrated and embedded in paraffin followed by sectioning and staining. Ashcroft scores were evaluated based on the results above. For IF and IHC staining, lung sections were stained with anti α-SMA, anti COL1A1, anti ProSPC, anti AQP5, anti vWF antibodies and anti-TIMP-1 antibodies. Measurements were taken from distinct samples.

### Determination of hydroxyproline

The HYP content in mouse lungs was determined using a hydrochloric acid hydrolysis kit according to the manufacturer's instructions. In brief, ~200 mg lung tissues were accurately weighed and hydrolyzed in hydrochloric acid solution at 100 °C for 2–6 h. After cooling, the hydrolysate was adjusted to a pH of 6.0–8.0 with NaOH and centrifuged at 16,000 rpm for 20 min. The collected supernatant was incubated with detection agents, and absorbance at 560 nm was measured to determine the HYP content using a microplate reader (Thermo Scientific, USA). Measurements were taken from distinct samples.

### Analysis of ROS in lung tissues

For flow cytometry, mice were intratracheally injected with 50 μL DCFH-DA (10 μM). After 20 min, the mouse lung tissues were collected and immediately ground into a single-cell suspension through a 40 μm filter. The process was operated on ice. Then, the level of ROS was quantified by flow cytometry. Measurements were taken from distinct samples.

### Determination of TIMP-1 and MMP-9 in mouse BALF

Mouse BALF was collected and centrifuged at $1000 \times g$ for 10 min at 4 °C to remove cells. The level of TIMP-1 and MMP-9 in mouse BALF of various groups were assayed using ELISA method as instructed by the manufacturer. Measurements were taken from distinct samples.

### Statistical analysis

All results are analyzed using GraphPad Prism software (version 8.0.2) and expressed as means ± SEM. Significant differences between two groups were analyzed by two-tailed unpaired Student's t test. Significant differences between three or more groups were analyzed by one-way ANOVA. ns, no significance, *P < 0.05, **P < 0.01, ***P < 0.001, and ****P < 0.0001. Differences were considered statistically significant at P < 0.05.

### Reporting summary

Further information on research design is available in the Nature Portfolio Reporting Summary linked to this article.

## Data availability

The RNA-seq data generated in this study have been deposited in the Genome Sequence Archive (https://ngdc.cncb.ac.cn/) under accession codes CRA033644. All data supporting the findings of this study are available within the article and its supplementary files. Any additional

requests for information can be directed to and will be fulfilled by the corresponding authors. Source data are provided with this paper.

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

## Acknowledgements

This work was supported by the Biological & Medical Engineering Core Facilities (Beijing Institute of Technology) for providing advanced equipment, the National Natural Science Foundation of China (22274011), Shenzhen Medical Research Funds (A2303070), Hebei Natural Science Foundation (B2024105013), the Beijing Institute of Technology Research Fund Program for Young Scholars (2020–2024), Beijing Natural Science Foundation-Haidian Original Innovation Joint Program (L242138), Beijing Chen Jumei Public Welfare Foundation (202524141003A), Basic and Applied Basic Research Foundation of Guangdong Province (2024A1515010775).

## Author contributions

C.L. and L.H. conceived and designed the study. C.L., G.L., and H.C. performed the experiments. G.L. collected and analyzed the clinical samples. C.W., Z.W., R.M., and S.L. helped with nanoplatform preparation and experimental assays. C.L. and H.C. facilitated the data and file processing. All authors discussed the results and commented on the

manuscript. C.L. wrote the manuscript. G.L. and L.H. further revised and edited the manuscript. All authors have read and approved the article.

## Competing interests

The authors declare no competing interests.
