## [Transparent Peer Review file · Nature Communications]

A Nanosystem Targeting Tissue Inhibitor of Metalloproteinase-1 for Continuous Spatiotemporal Idiopathic Pulmonary Fibrosis Therapy

Corresponding Author: Dr Lili Huang

Version 0:

Reviewer comments:

Reviewer #1

(Remarks to the Author)

The manuscript utilizes a ROS-responsive nanosystem targeting TIMP-1 to regulate ECM metabolic balance in fibrosis, presenting an innovative therapeutic strategy. By simultaneously scavenging ROS and releasing therapeutic exosomes, it offers a dual therapeutic mechanism for IPF. The in vivo and in vitro experimental data are abundant, and the significant antifibrotic effects of the nanosystem have been fully validated in a mouse model.

However, there are some issues that should be addressed:

1. Line 141—"When the ratio of Mexo to aT was 5:1 (w/w), the Mexo-cl-aT displayed the greatest antifibrotic effect on transforming growth factor- β 1 (TGF- β 1)-induced primary mouse lung fibroblasts (MLFs)"
In this case, what is the cytotoxic effect of Mexo-cl-aT on cells? Was the CCK8 assay used for detection? Is this screening ratio method reasonable? Should the effect on cells be prioritized? Additionally, how was the concentration of TGF- β determined?
2. How does Mexo-cl-aT achieve better collagen degradation? Is it through an extracellular or intracellular degradation pathway?
3. In in vivo studies, only the effect of a single dose administration was reported, but the effect of different doses of Mexo-cl-aT on IPF was not explored in depth. Dose optimisation experiments are recommended to determine the most effective and safest dose range. There are also fewer time-dependent studies, and research could add monitoring of drug effects at different time points to determine the optimal timing of therapeutic effects.
In addition, IPF is a chronic progressive disease and only one time point (10 days) was observed in the manuscript, it is recommended to extend the observation period to determine the presence of delayed side effects and cumulative organ toxicity.
4. The current experimental control groups, which mainly consist of PBS, aT, Mexo and Mexo-fl-aT group, lack a direct comparison with other standard antifibrotic treatments such as pirfenidone and nintedanib. The addition of a positive control group for these drugs could provide a clearer picture of the advantages and disadvantages of Mexo-cl-aT over existing treatments.
5. Figure 6C—Most models of fibrosis are quite patchy in nature, thus just choosing one relatively high powered microscope field is not sufficient to quantify the levels of fibrosis. The addition of mRNA and western blotting results is recommended to more fully quantify the therapeutic effect of Mexo-cl-aT.

Reviewer #2

(Remarks to the Author)

The manuscript by Li et al. presents an antibody-based therapeutic approach for advanced lung fibrosis. While the study touches on an important issue, the work needs further experimental evidence before can be consider for publication in NC:

1. Choice of TIMP-1 as a therapeutic target. The justification for selecting TIMP-1 as the target is not sufficiently supported. Although increased expression of TIMP-1 is observed in fibrotic areas, this alone does not establish it as a viable therapeutic target. Moreover, the efficacy of inhibiting TIMP-1—or any single-factor inhibition, particularly in advanced lung fibrosis—remains controversial in the previous studies.

2. Boronic acid-based ROS-responsive materials. The materials described are not broadly responsive to ROS but are, in fact, solely responsive to hydrogen peroxide at relatively high concentrations (micromolar to millimolar range). Achieving such concentrations in the human body is challenging, which often necessitates the use of GOx to induce H₂O₂ production in previous studies. Furthermore, based on the authors' results, the nanosystem did not demonstrate significantly enhanced sensitivity to hydrogen peroxide (100 μ M), raising concerns about whether it will function as designed in vivo or whether the observed effects are actually obtained from a different mechanism.

3. Rationale in the use of exosomes. The inclusion of exosomes in this study needs further consideration. Given that polymer conjugation can yield similar outcomes in terms of lung accumulation and hydrogen peroxide scavenging when administered via inhalation, the added value of exosomes in this context is not clear.

4. Mechanistic study. The mechanistic investigation needs further depth experiments to get to certain mechanistic claim. For example, the authors suggest that TIMP-1 inhibition restores MMP activity, but MMP is a broad family with diverse roles depending on the subtype. Moreover, only MMP-9 was evaluated in the study, whose role in lung fibrosis is both controversial and largely unknown, therefore it would be even beneficial to explore whether MMP-9 can be adopted as a robust pathogenesis for lung fibrosis. Additionally, other experiments, such as measuring ROS content and cell viability, seem more indicative of particle functionality rather than offering insight into the underlying mechanisms.

Reviewer #3

(Remarks to the Author)

In this manuscript, Li et al. developed a nano system targeting tissue inhibitor of metalloproteinase-1 (TIMP-1) for continuous IPF therapy. It seems that the anti-fibrotic effects were good, but more data or evidence are needed to confirm the anti-fibrotic effects without adverse effects. I have several concerns with the claimed mechanism of anti-fibrosis.

1. I would concern about the mechanism of the anti-fibrotic effects. The authors claimed that the nano-system with anti-TIMP-1 could rebalance TIMP-1 and MMP9 further ameliorated BLM-induced fibrosis. Based on the published articles, the claim was still controversial. From their claim, the anti-TIMP-1 treatment would upregulate MMP9 and dissolve the fibrosis. However, the increased MMP9 would enhance inflammatory cells infiltration and led to increased inflammation. Besides, the increased MMP9 would lead to increased TGF-beta activation that further aggravate the fibrosis. I highly recommended the authors provided more information about the phenotype of in vivo anti-fibrosis effects.

2. Figure 1. The author correlated TIMP-1 with fibrosis in patients and fibrotic mice. However, I am much curious about the relation of MMP9 and Fibrosis. If there is an imbalance with TIMP-1 and MMP9, whether MMP9 would decrease within the fibrotic lung?

3. Authors focused on fibroblasts and epithelial with in vitro experiments. While the in vivo experiments were much more complicated than the in vitro system. The author should provide more information about the cellular source of TIMP1 and MMP9, which would smooth the results to fibroblasts and epithelial. Besides, whether the anti-TIMP-1 treatment would increase the inflammatory cell infiltration into the lung?

4. Figure 6. I am quite curious about the effects when the nano system applied to the mice in vivo. Though the authors provide information about the anti-fibrotic effect, I will ask that what the effects that would be about the inflammation. Some related cytokines IL-1b, TNF-a, IL-6 and the most critical TGF-b expression should be detected to fulfill its anti-fibrotic effects.

5. Figure 6. As for the Masson's and H&E staining, authors are expected to provide full lung lobe images with zoomed area for the readers better to evaluate the anti-fibrotic effects. The fibrosis in the lung were very heterogeneity. It would be better for the readers to evaluated the anti-fibrosis effects. Besides, IF staining to α -SMA and COL-1 were bad quality, confocal images would be better if the author could do that.

6. Figure 6. The levels of TIMP1 and MMP9 in the BALF were not that convincing, since majority of the cytokines were in the lung interstitial. Also, there were huge variation when BAL was prepared. Western blot detecting the two important proteins with the lung tissue are needed to better illuminate the protein levels in the lung. The results would better illuminate the balance between the two proteins.

7. BLM induced pulmonary fibrosis would dissolve within 3-5 weeks after BML treated. If the author could provide more information about the Nano system against Crystalline silica induced fibrosis would be more attracting. Since CS particle would induced persist and progressive pulmonary fibrosis.

8. How many mice were used in the experiment of each group. HE Masson staining would be evaluated with semi quantification.

9. The anti-fibrotic effects would rely more on the anti-ROS system or the anti-TIMP-1 effects?

Reviewer #4

(Remarks to the Author)

Reviewer #5

(Remarks to the Author)

Overall:

This manuscript describes a study deploying umbilical cord-derived mesenchymal exosomes decorated with ROS-cleavable antibodies to TIMP-1 and delivered intratracheally as a potential therapeutic nanosystem for idiopathic pulmonary fibrosis. While many aspects of the study are intriguing, the exact cellular and molecular targets of this complex nanosystem are poorly defined, and the data as presented do not clearly reveal therapeutic targets and mechanism of action. The use of a single model which does not recapitulate many aspects of the human disease limits the broad applicability of the findings.

Major Points:

- The rationale for the focus on TIMP-1 and MMP9 specifically is poorly defined. Additional discussion is needed about which MMPs are implicated in IPF pathogenesis and why TIMP-1 was chosen (ie does it have higher affinity binding for these specific MMPs?).
 - Figure 1 - TIMP staining – Figures are low resolution and lack use of additional cell markers (Krt5, aSMA etc) to determine location of TIMP expression. In human IPF images shown the staining is rather diffuse. Is it intracellular or extracellular? Their BAL ELISA would suggest extracellular, but their staining shows it in tissue - is it being produced / secreted by specific cell types in the IPF lung? Can the authors use mouse lung tissue from their bleomycin experiment to look for TIMP levels via ELISA or Western blot?
 - Figure 1g/h – the method used to stain for “ROS” is not specified in the methods section. What was used? Very few reagents will react with all forms of ROS and are not stoichiometric. DCFH-DA was used later in the paper and may have been used here? If so, DCFH-DA is not a reliable measure of all ROS and has several limitations (e.g., PMID: 22027063). Additionally, staining fixed tissue sections for ROS is not ideal – reactive species are produced and reduced dynamically and fixation will not preserve these species. The authors should state these limitations clearly and use a complementary assay to assess redox dysregulation in these samples, such as the formation of oxidative byproducts or modifications (see PMID: 29988126). It is also unclear if human lung tissue was formalin fixed or frozen, as well as any demographic information or source of IPF and healthy control lung tissues - these should also be specified in the methods.
 - Rationale for choice of mesenchymal exosomes (e.g., vs. an engineered nanoparticle) could be better developed. Is the goal to protect the antibody, or to only release it when ROS levels are high? Or is exosome delivery meant to reach intracellular space? Also, what is the rationale for sourcing exosomes from umbilical cord mesenchymal stem cells, and what additional molecular do these exosome preparations contain?
 - Figure 3 – The authors treat MLF stimulated with TGFB with the exosome/TIMP antibody, but do not confirm that TIMP and/or MMP activity/levels were affected by TGFB treatment in these fibroblasts. Staining for TIMP/MMP or an activity assay in the TGFB treated fibroblasts would be helpful here. Also, in Fig3a, there is no information on number of fields of view assessed or replicates, or any quantification for these samples. It is hard to get an idea of the scale of the change from isolated images. These are cells in culture, so it should be possible to also perform western blotting and/or ELISAs of the targets investigated in this panel (aSMA, col1a1, and add TIMP)?
 - Figure 3 – when MLF are treated with the exosome alone, there is significant reduction of aSMA and it looks like collagen (although it is not clear if this comparison was not significant or just not reported). What is the proposed explanation for this? It looks like the exosome has equal effect to the antibody alone? In the scheme in Figure 1 it shows the proposed effect of Mexo alone acting on vascular endothelial cells and epithelial cells, but this point is not really discussed in the introduction or results.
 - Figure 4 in general – it seems like the Mexo-fl-aT and the Mexo-cl-aT both have the same effect on viability, apoptosis, and wound healing, with the only difference being that Mexo-fl-aT is not able to increase MMP9 activity. Would this indicate that these phenotypic effects on wound healing, apoptosis, cell viability, are not mediated through MMP9 activity? For some of the endpoints, like wound healing, it looks like the exosome alone increases the response.
 - Figure 4f - Again, DCFH-DA is not a reliable probe for ROS generation. Something like Amplex Red might be a better assay here, which also focuses on extracellular H2O2 levels, and would better reflect the extracellular oxidative environment in which the Mexo-cl-aT would be cleaved.
 - Figure 6g – TIMP1 levels decrease in response to these treatments, but even the Mexo alone condition causes this. What is the proposed mechanism for this? Does binding of TIMP-1 with the aT antibody cause it to be undetected by the ELISA? Also, this is in BAL- do tissue levels of TIMP-1 decrease as well? Can ELISA/staining/WB be performed on these?
 - Can the authors demonstrate that when aT is released from Mexo-cl-aT, that the antibody then binds TIMP? Can this complex be identified and measured, ie via immunoprecipitation from in vivo samples? This would also help determine non-specific binding of aT to other targets (other TIMPs?) occurs in vivo.
 - The discussion is not very detailed and the first two paragraphs merely summarize the results of the study, while only the last one discusses broader implications. What are the limitations of this study?
 - Overall, the main goal of the treatment is unclear – is the Mexo-cl-aT meant to deliver the anti-TIMP antibody and scavenge ROS through cleavage of the linker? Or is the Mexo component serving some value in wound repair? I do not think this distinction is well described in the introduction/results sections. In lines 341 – 351 of the discussion this is mentioned, although the statement “Mexo have a natural ability to home to injury sites” is not cited. A more thorough introduction about why Mexo was chosen and what is known about exosomes ability to promote wound repair is needed in the introduction.
- Minor Points:
- Citations appear to be after the period for each sentence.

- Methods – “Determination of ROS in lung tissues” and “Determination of TIMP-1 in BALF” have the same description, which is for the TIMP measurement. ROS method section should be updated.
- Figure 2h – How was H₂O₂ quantified in these samples? This is not stated in the methods section.
- The MMP9 activity assay in Figure 4B makes more sense to be included in Figure 3 – it is from the fibroblast experiment and involves the same treatments. The y-axis of this figure is confusing – is this activity or abundance (the ng/mL suggests abundance). This is also an issue in Figure 6C.
- Figure 4c – it is unclear what the treatment is. I’m assuming it is the bleomycin treatment, but no label is provided on the figure and no information on dose is given in the legend. The significance notation here is also unclear – is everything significant, or is it referring to a specific comparison?
- Figure 4d – I think having the graph shown in Figure S7a would be helpful here to compare the apoptotic cells in the same format as other assays
- Mouse model is referred to as “IPF” – these mice don’t have IPF, they’ve been injured with bleomycin

Reviewer #6

(Remarks to the Author)

Version 1:

Reviewer comments:

Reviewer #1

(Remarks to the Author)

Even though the author added RNA sequencing to evaluate the therapeutic effect of Mexo-cl-aT on pulmonary fibrosis, this only reflects changes at the genetic level, and the alterations in fibrosis marker proteins remain unknown. Additionally, could you provide the uncropped original membrane for Fig. 2A?

Reviewer #2

(Remarks to the Author)

Authors have addressed all the issues raised by this reviewer.

Reviewer #3

(Remarks to the Author)

The authors conducted substantial experiments to prove their conclusions. However, I still have several major concerns.

1. The first concern is about the cellular source of MMP9. Though there was report indicated that Fibroblasts are recognized as one of the cellular sources of MMP-9 (PMID:36352225), it was not in the fibrotic lung micro-environment. Still, the major cellular source of MMP9 is leucocytes, especially Macrophages. The author claimed that delivering Mexo-cl-aT would decrease CD45+ cells, as well as other pro-inflammatory cytokines. In that case, how the MMPs increased, what is the cellular source. We suppose that the MMPs came from fibroblast, the alleviated fibrosis did not support the hypothesis that it came from fibroblasts. The cellular source of increased MMPs should be explored in the circumstance. Extend experimental data is needed to support authors conclusion and for self-justification.

Overall, the decreased cytokine and CD45+ cells and fibroblast make me confused about how MMP9 increased in the lung microenvironment.

2. The author believed the effects of the Mexo-cl-aT therapy depends on the anti-TIMP1 activity. However, the author did not intervene the TIMP1 in BLM-treated mice. In the Timp1 knockout mice, whether it could alleviate BLM-induced pulmonary fibrosis. If so, as reported in the previous reports, the most important issue is that whether Mexo-cl-aT treatment is still effective in the Timp1 knockout mice. This experiment can answer whether the treatment depends on the effect of TIMP1.

3. Why is there was significant differences in the results between the positive control Pirfenidone and the treatment with Mexo-cl-aT? (FigS18) Why there were 8 groups in the elisa experiments, but 7 groups in the Immunofluorescence staining of CD45+ cells in mouse lung sections.

4. ELISA analysis of TIMP-1 and MMP-9 would not evaluate the activity of TIMP1 and MMP9. It can only evaluate the protein contents. Gelatin enzyme spectrum experiment would demonstrate the enzyme activity.

5. In FigS1, it seems that TIMP1 did not come from a-SMA+ fibroblasts. If the authors stained macrophages markers F4/80, it may be more co-localization.

6. The time points of Mexo-cl-aT intervention in vivo is still lack experimental evidence in Fig 6a. Confirmed in Fig S18, Mexo-cl-aT could exert anti-inflammatory efficacy during inflammatory stage. In addition, the ROS generation could also be

observed in inflammatory stage. Please provide experiment data about Mexo-cl-aT intervention including inflammatory stage and fibrotic stage in IPF model, which could make the '10 day' time point rational.

7. According to the Graphical Abstract, the core anti-fibrosis target of Mexo-cl-aT was regulating the balance between TIMP1 and active MMP9. However, the co-relation of Mexo-cl-aT and TIMP1 should be confirmed by TIMP1 encoding gene knockout mice, and the relationship between active MMP9 and TIMP1 also need demonstration. Following the Transcriptome analysis in Fig S19, Mexo-cl-aT treatment increased the expression of massive MMP family genes, which was contradictory against TIMP1-MMP9 balance. Thus, the 'Mexo-cl-aT-TIMP1-MMP9-anti-fibrosis' pathway could not be established with compelling evidence.

8. Although the authors weakened the content about MMP9 in the revised manuscript, the relationship between active MMP9 and reduced fibrosis in IPF should not be explained so simple, which was mentioned in my previous Q2. Please add some test index into in vivo experiment, which can better reflect upregulated collagen degradation specific governed by active MMP9.

9. In Fig S17, the pirfenidone treatment could not effectively hamper lung fibrosis in large field view, but TGF- β and TIMP1 reduced collectively. What's more, the level of IL-6 and TNF- α should be noticed in pirfenidone treated group in Fig S18.

Reviewer #4

(Remarks to the Author)

Reviewer #5

(Remarks to the Author)

Overall: The authors have responded to the reviewer concerns, and have provided extensive additional data and figures which have provided additional context and greatly improved the impact of the work.

Major Comments: none

Minor Comments:

1. Lines 67-72: Thank you for adding this helpful text. It is added in a paragraph discussing ROS, and it seems from the data that the phenylboronic ester bond, not the Mexo themselves, are what is scavenging ROS in this system. Please make the added text a separate paragraph.
2. Fig. 4.g.: it would be helpful to add reference lines for "no migration" (PBS condition) to all the panels so that the comparable effects of treatments can be more easily seen. Also for Fig. S10c
3. Line 307: "In a comparative study that pirfenidone was as the positive control"; suggest change to "In a comparative study in which pirfenidone was used as a positive control" to improve clarity.
4. Line 403: "of exosomes, which are persistent issues in bio-optimization"; suggest change to "of exosomes-persistent issues in bio-optimization" for clarity.
5. Line 404: "Additionally, the antifibrotic potential"; suggest change to "Furthermore, the approach may not be optimized for all forms of fibrosis. For example, the antifibrotic potential" for clarity.

Reviewer #6

(Remarks to the Author)

Version 2:

Reviewer comments:

Reviewer #1

(Remarks to the Author)

Reviewer #3

(Remarks to the Author)

Thanks for the authors response. I have no further questions. The manuscript could be accepted for publication.

Reviewer #4

(Remarks to the Author)

Response to Referees

Manuscript entitled “A Nanosystem Targeting Tissue Inhibitor of Metalloproteinase-1 for Continuous Spatiotemporal Idiopathic Pulmonary Fibrosis Therapy ” (NCOMMS-24-53302).

Responses to the comments from the reviewers (Note: All the responses here are in blue. All the changes have been highlighted in red in the revised manuscript).

Reviewer #1 (Remarks to the Author):

The manuscript utilizes a ROS-responsive nanosystem targeting TIMP-1 to regulate ECM metabolic balance in fibrosis, presenting an innovative therapeutic strategy. By simultaneously scavenging ROS and releasing therapeutic exosomes, it offers a dual therapeutic mechanism for IPF. The in vivo and in vitro experimental data are abundant, and the significant antifibrotic effects of the nanosystem have been fully validated in a mouse model.

However, there are some issues that should be addressed:

- **Response:** We thank the reviewer for the positive comments and for recognizing our work as an “**innovative therapeutic strategy**”.

1. Line 141-“When the ratio of Mexo to aT was 5:1 (w/w), the Mexo-cl-aT displayed the greatest antifibrotic effect on transforming growth factor- β 1 (TGF- β 1)-induced primary mouse lung fibroblasts (MLFs)” In this case, what is the cytotoxic effect of Mexo-cl-aT on cells? Was the CCK8 assay used for detection? Is this screening ratio method reasonable? Should the effect on cells be prioritized? Additionally, how was the concentration of TGF- β determined?

- **Response:** Thanks for the reviewer’s kind reminder. we evaluated the cytotoxic effect of Mexo-cl-aT on cells by CCK8 assay. The results showed that 5 μ g/mL Mexo + 1 μ g/mL aT (Mexo:aT=5:1 (w/w)) displayed no significant toxicity or side effects on A549, HUVEC, and MLFs. We have added the data as **Fig.S7** and

corresponding description (line 149-150) in the revised manuscript.

Fig.S7 The cytotoxic effect of Mexo-cl-aT on A549, HUVEC, and MLF cells by CCK8 assay. The cells were treated with different ratio of Mexo and aT for 24 hours before measurement (N=3). Data are means \pm SEM; ns, not significant (one-way ANOVA with Sidak's multiple comparisons test).

- **Response:** The concentration of 10 ng/mL TGF- β 1 employed in our study was selected based on well-established protocols from previous studies (PMID: 36402779; PMID: 35731866; PMID: 39791575) demonstrating its efficacy in activating fibroblasts *in vitro*.

2. How does Mexo-cl-aT achieve better collagen degradation? Is it through an extracellular or intracellular degradation pathway?

- **Response:** We are sorry for the confusion. When Mexo-cl-aT enters a fibrotic microenvironment characterized by high levels of ROS, its phenylboronic acid ester bond breaks, releasing aT. Since MMP-9 (<https://www.uniprot.org/uniprotkb/P41245/entry>) and TIMP-1 (<https://www.uniprot.org/uniprotkb/P12032/entry>) are secreted proteins that reside in the extracellular matrix, where aT can effectively exert its function. Specifically, aT neutralize free TIMP-1, leading to increased activity of MMP-9. As shown in **Fig. 4b-c**, Mexo-cl-aT treatment resulted in significant decrease of free TIMP-1 concentration and increase of MMP-9 activity in the culture supernatant of TGF- β 1-stimulated mouse lung fibroblast (MLF) cells. We have added the data as **Fig. 4b-c** and corresponding description (line 200-204) in the

Fig. 4b-c. The levels of TIMP-1 (PBS vs. Mexo-cl-aT, $P < 0.0001$) and active MMP-9 (PBS vs. Mexo-cl-aT, $P = 0.0077$) in MLF culture supernatants ($N=3$). Data are means \pm SEM; ns, not significant; * $P < 0.05$; ** $P < 0.01$; *** $P < 0.001$; **** $P < 0.0001$ (one-way ANOVA with Dunnett's multiple comparisons test).

3. In in vivo studies, only the effect of a single dose administration was reported, but the effect of different doses of Mexo-cl-aT on IPF was not explored in depth. Dose optimisation experiments are recommended to determine the most effective and safest dose range. There are also fewer time-dependent studies, and research could add monitoring of drug effects at different time points to determine the optimal timing of therapeutic effects. In addition, IPF is a chronic progressive disease and only one time point (10 days) was observed in the manuscript, it is recommended to extend the observation period to determine the presence of delayed side effects and cumulative organ toxicity.

- **Response:** Much thanks to the reviewers for the in-depth review of our manuscript. According to the reviewer's suggestion, we performed the dose optimization experiments to determine the most effective and safest dose range. Simultaneously, we extended the observation period from 18 days to 28 days to determine the presence of delayed side effects and cumulative organ toxicity. In addition, we monitored the drug effects at different time points via lung function testing to determine the optimal timing of therapeutic effects.

- The dosage and frequency of Mexo-cl-aT administration. Dose 1, high-dose single administration of Mexo-cl-aT, contains 10 µg Mexo and 2 µg aT; Dose 2, low-dose multiple administrations. The total dose is the same as Dose 1; Dose 3, high-dose multiple administrations. The total dose is three times that of Dose 1 and Dose 2.
- According to **Fig. S16a**, three different doses of Mexo-cl-aT were individually administered to a bleomycin-induced fibrosis mouse model via inhalation, and lung function was measured at 10, 18, and 28 days post-bleomycin administration. The results showed that compared to the control group, the breathing frequency of the mice in the PBS group was significantly enhanced, while the levels of peak expiratory flow rate and tidal volume were reduced significantly. Compared with the PBS group, all the lung function indexes were restored considerably in three different doses of Mexo-cl-aT groups (**Fig. S16b-d**). Obviously, Dose 3 (high-dose multiple administrations) is less effective than Dose 1 (high-dose single administration) and Dose 2 (low-dose multiple administrations) in restoring respiratory frequency and tidal volume in mice with pulmonary fibrosis (**Fig. S16b-d**). Moreover, multiple administrations may impose a burden on the respiratory function of the mice. Therefore, Dose 1 is most beneficial for restoring normal lung function in mice with pulmonary fibrosis. We have added the data as **Fig.S16** and corresponding description (line 301-303) in the revised manuscript.

Fig. S16 Inhalation of Mexo-cl-aT restores the pulmonary functions of mice with bleomycin-induced pulmonary fibrosis. **a.** Illustration of animal experiment procedure. **b-d.** Measurements of pulmonary function parameters including frequency (b), peak expiratory flow rate (c), and tidal volume (d) at day 10, 18, 28 treatment (N = 6). Data are means \pm SEM. Dose 1, high-dose single administration of Mexo-cl-aT (total equivalent: 10 μ g Mexo and 2 μ g aT). Dose 2, low-dose multiple

administrations of Mexo-cl-aT (total equivalent: 10 μ g Mexo and 2 μ g aT). Dose 3, high-dose multiple administrations of Mexo-cl-aT (total equivalent: 30 μ g Mexo and 6 μ g aT). ns, not significant; *P < 0.05; **P < 0.01; ***P < 0.001; ****P < 0.0001 (one-way ANOVA with Dunnett's multiple comparisons test).

- To further assess the therapeutic efficacy of three different doses of Mexo-cl-aT, we used HE and Masson staining to observe the histopathological changes of lung tissues in the different groups. The control group showed normal lung structure without inflammation, while the PBS group exhibited severe lung damage and inflammation. Mexo-cl-aT treatment significantly improved lung histopathology, reduced alveolar wall damage, and had antifibrotic effects compared to the PBS group (**Fig. S15**). Particularly noteworthy was the superior antifibrotic performance of the Dose 1 group, evidenced by a cleared and intact alveolar structure and diminished collagen deposition (**Fig. S15**). Taken together, Dose 1 group could effectively improve lung function, and reduce tissue fibrosis in mice with pulmonary fibrosis. We have added the data as **Fig. S15** and corresponding description (line 301-303) in the revised manuscript.

Fig. S15 Anti-fibrotic effects of Mexo-cl-aT inhalation treatment in mice of bleomycin-induced pulmonary fibrosis. a. Illustration of animal experiment procedure. **b.** H&E and Masson staining of mouse lungs after different treatments (N=6). Dose 1, high-dose single administration of Mexo-cl-aT (total equivalent: 10 μg Mexo and 2 μg aT). Dose 2, low-dose multiple administrations of Mexo-cl-aT (total equivalent: 10 μg Mexo and 2 μg aT). Dose 3, high-dose multiple administrations of Mexo-cl-aT (total equivalent: 30 μg Mexo and 6 μg aT). Scale bar = 200 μm .

- Finally, we assessed the biosafety of Mexo-cl-aT *via* hematologic and biochemical analysis. The hepatic and renal function parameters remained within normal reference ranges; however, treatment with Dose 3 resulted in a significant decrease in BUN (**Fig. S20**). In summary, The Dose 1 employed in our study is the most effective and safest dose range. We have added the data as **Fig. S20** and corresponding description (line 334-336) in the revised manuscript.

Fig. S20 Biosafety of Mexo-cl-aT inhalation treatment in a bleomycin-induced pulmonary fibrosis mouse model. a. Illustration of animal experiment procedure. **b.**

Evaluation of liver and kidney functions in mice treated with different doses of Mexo-cl-aT (N=6). Dose 1, high-dose single administration of Mexo-cl-aT (total equivalent: 10 µg Mexo and 2 µg aT). Dose 2, low-dose multiple administrations of Mexo-cl-aT (total equivalent: 10 µg Mexo and 2 µg aT). Dose 3, high-dose multiple administrations of Mexo-cl-aT (total equivalent: 30 µg Mexo and 6 µg aT). ns, not significant; *P < 0.05 (one-way ANOVA with Dunnett's multiple comparisons test).

4. The current experimental control groups, which mainly consist of PBS, aT, Mexo and Mexo-fl-aT group, lack a direct comparison with other standard antifibrotic treatments such as pirfenidone and nidaneb. The addition of a positive control group for these drugs could provide a clearer picture of the advantages and disadvantages of Mexo-cl-aT over existing treatments.

- **Response:** Thank the reviewer for the suggestion. We added pirfenidone, a standard antifibrotic treatment, as a positive control and evaluated its therapeutic efficacy using HE and Masson staining. The control group showed normal lung structure, while the PBS group exhibited severe lung damage. Mexo-cl-aT and pirfenidone treatment significantly improved lung histopathology, reduced alveolar wall damage. Remarkably, the Mexo-cl-aT group demonstrated better antifibrotic effects compared to the pirfenidone group, as shown by a well-preserved alveolar structure and reduced collagen accumulation (**Fig. S17a**). We have added the data as **Fig. S17a** and corresponding description (line 304-305) in the revised manuscript.

Fig. S17 Anti-fibrosis effect of Mexo-cl-aT vs. pirfenidone in a bleomycin-induced pulmonary fibrosis mouse model. a. HE and masson staining of the lung sections of mice treated with different formulations. Scale bar = 200 μ m.

5. Figure 6C-Most models of fibrosis are quite patchy in nature, thus just choosing one relatively high powered microscope field is not sufficient to quantify the levels of fibrosis. The addition of mRNA and western blotting results is recommended to more fully quantify the therapeutic effect of Mexo-cl-aT.

- **Response:** Thank the reviewer for the good suggestion. We have added full lung lobe images with zoomed area in **Fig. 6c** of the revised manuscript.
- **Response:** We further evaluated the therapeutic effect of Mexo-cl-aT using RNA sequencing. The results revealed upregulated expression of MMP-related genes, including MMP1b, MMP3, MMP7 MMP8 MMP9 and MMP12, in lungs treated with Mexo-cl-aT. Furthermore, the upregulation of MMPs was significantly higher than that of TIMPs, indicating a shift towards extracellular matrix

degradation. Additionally, the expression of genes associated with oxidative stress decreased, suggesting effective alleviation of lung tissue damage. We have added the data as **Fig. S19** and corresponding description (line 317-320 and line 324-327) in the revised manuscript.

Fig. 6c. Masson staining of mouse lung sections with different treatments. Scale bar = 200 μm .

Fig. S19 Transcriptome analysis in the lung of mice with bleomycin-induced pulmonary fibrosis. **a.** Heatmap of MMP- and TIMP-related genes. **b.** Heatmap of oxidative stress-related genes. N=3. M-c-a: Mexo-cl-aT.

Reviewer #2 (Remarks to the Author):

The manuscript by Li et al. presents an antibody-based therapeutic approach for advanced lung fibrosis. While the study touches on an important issue, the work needs further experimental evidence before can be consider for publication in NC:

- **Response:** We appreciate the reviewer's insightful comments. In response, we have conducted additional relevant experiments and implemented appropriate revisions to enhance the manuscript.

1. Choice of TIMP-1 as a therapeutic target. The justification for selecting TIMP-1 as the target is not sufficiently supported. Although increased expression of TIMP-1 is observed in fibrotic areas, this alone does not establish it as a viable therapeutic target. Moreover, the efficacy of inhibiting TIMP-1—or any single-factor inhibition, particularly in advanced lung fibrosis—remains controversial in the previous studies.

- **Response:** Thanks for the reviewer's kind reminder. Previous studies have reported that TIMP-1 is overexpressed by activated fibroblasts and promotes fibrosis by inhibiting MMP-mediated ECM degradation (PMID: 30243515; PMID35715541), and the impaired proteases and anti-proteases balance was found in IPF (PMID: 29518524). We have preliminarily verified the effectiveness of anti-TIMP-1 therapy to enhance collagen degradation *in vitro*, as shown in **Fig. 3a**. COL1A1 was upregulated in MLFs after TGF- β 1 stimulation. Inhibiting TIMP-1 by aT promoted collagen degradation, and finally decreased COL1A1 (weakened red fluorescence signal).

Fig. 3a IF staining of α -SMA and COL1A1 in MLFs.

- **Response:** Mexo-cl-aT does more than just single-factor inhibition to combat fibrosis. It contains three active ingredients: Mexo inhibits fibroblast activation and repairs epithelium and endothelium; phenylboronic acid ester-based ROS-cleavable linker (cl) removes excess H₂O₂ at the lesion; aT restores MMP activity to promote collagen degradation. They synergize to achieve superior anti-fibrotic treatment.

2. Boronic acid-based ROS-responsive materials. The materials described are not broadly responsive to ROS but are, in fact, solely responsive to hydrogen peroxide at relatively high concentrations (micromolar to millimolar range). Achieving such concentrations in the human body is challenging, which often necessitates the use of GOx to induce H₂O₂ production in previous studies. Furthermore, based on the authors' results, the nanosystem did not demonstrate significantly enhanced sensitivity to hydrogen peroxide (100 μ M), raising concerns about whether it will function as designed in vivo or whether the observed effects are actually obtained from a different mechanism.

- **Response:** We thank the reviewer for drawing this point to our attention. We measured the concentration of hydrogen peroxide in lung tissues of

bleomycin-induced pulmonary fibrosis mouse model, and the concentration reached 0.9 mM (**Fig. S13**). It is feasible that our nanosystem respond to the pulmonary fibrosis microenvironment. Furthermore, extensive research has demonstrated that the lesion site of pulmonary fibrosis is characterized by a persistent and aberrant production of reactive oxygen species, including hydrogen peroxide, which leads to the pathophysiology of the disease (PMID: 33201251; PMID: 37707699). We have added the data as **Fig. S13** and corresponding description (line 268) and experimental procedure (line 597-605) in the revised manuscript.

Fig. S13 The level of H₂O₂ in lung tissues of bleomycin-induced pulmonary fibrosis mouse model. P = 0.0181, N=3. Data are means ± SEM. *P < 0.05 (two-tailed unpaired Student's *t* test).

3. Rationale in the use of exosomes. The inclusion of exosomes in this study needs further consideration. Given that polymer conjugation can yield similar outcomes in terms of lung accumulation and hydrogen peroxide scavenging when administered via inhalation, the added value of exosomes in this context is not clear.

- **Response:** Thanks to the reviewer's suggestion. MSC-derived exosomes (Mexo) are ideal nanocarriers for lung-targeted drug delivery via inhalation, given their optimal size (30–150 nm), biocompatibility, low immunogenicity, and inherent tropism for injured or inflamed tissues, offering advantages over polymer nanoparticles. Furthermore, they can transfer bioactive molecules (miRNAs, proteins, lipids) to recipient cells, exerting diverse therapeutic effects such as enhancing angiogenesis, reducing fibrosis, and stimulating cell proliferation and

differentiation. Additionally, Mexo exhibit potent immunomodulatory properties by suppressing pro-inflammatory cytokines and promoting anti-inflammatory responses, emerging as a promising cell-free therapy for regenerative medicine, inflammation-related diseases, and organ injury (PMID: 32855385; PMID: 32939235). In our manuscript, we also demonstrate that Mexo serves not only as a delivery system but also as an active therapeutic ingredient, inhibiting fibroblast activation (**Fig. 3a**) and repairing the epithelium (**Figs. 4d-g, S9**) and endothelium (**Fig. S10**) in the treatment of pulmonary fibrosis. In the introduction of the revised manuscript (line 67-72), we have provided a comprehensive explanation for the selection of Mexo.

4. Mechanistic study. The mechanistic investigation needs further depth experiments to get to certain mechanistic claim. For example, the authors suggest that TIMP-1 inhibition restores MMP activity, but MMP is a broad family with diverse roles depending on the subtype. Moreover, only MMP-9 was evaluated in the study, whose role in lung fibrosis is both controversial and largely unknown, therefore it would be even beneficial to explore whether MMP-9 can be adopted as a robust pathogenesis for lung fibrosis. Additionally, other experiments, such as measuring ROS content and cell viability, seem more indicative of particle functionality rather than offering insight into the underlying mechanisms.

- **Response:** We sincerely thank the reviewers for the in-depth review of our manuscript. We further evaluated the antifibrotic effect of Mexo-cl-aT through transcriptome analysis. The results showed that mice treated with Mexo-cl-aT showed the upregulated expression of MMP-related genes, including MMP-1, MMP-2, MMP-3, MMP-7, MMP-8, MMP-9 and MMP-12. Furthermore, the upregulation of MMPs was significantly higher than that of TIMPs, indicating a shift towards extracellular matrix degradation. Additionally, the expression of genes associated with oxidative stress decreased, suggesting effective alleviation of lung tissue damage. We have added the data as **Fig. S19** and corresponding description (line 317-320 and line 324-327) in the revised manuscript.

Fig. S19 Transcriptome analysis in the lung of mice with bleomycin-induced pulmonary fibrosis. **a.** Heatmap of MMP- and TIMP-related genes. **b.** Heatmap of oxidative stress-related genes. N=3. M-c-a: Mexo-cl-aT.

Reviewer #3 (Remarks to the Author):

In this manuscript, Li et al. developed a nano system targeting tissue inhibitor of metalloproteinase-1 (TIMP-1) for continuous IPF therapy. It seems that the anti-fibrotic effects were good, but more data or evidence are needed to confirm the anti-fibrotic effects without adverse effects. I have several concerns with the claimed mechanism of anti-fibrosis.

- **Response:** We sincerely appreciate the reviewer's insightful comments. Below we have conducted additional relevant experiments and implemented appropriate revisions to enhance the manuscript.

1. I would concern about the mechanism of the anti-fibrotic effects. The authors claimed that the nano-system with anti-TIMP-1 could rebalance TIMP-1 and MMP9 further ameliorated BLM-induced fibrosis. Based on the published articles, the claim was still controversial. From their claim, the anti-TIMP-1 treatment would upregulate MMP9 and dissolve the fibrosis. However, the increased MMP9 would enhance inflammatory cells infiltration and led to increased inflammation. Besides, the increased MMP9 would lead to increased TGF-beta activation that further aggravate the fibrosis. I highly recommended the authors provided more information about the phenotype of in vivo anti-fibrosis effects.

- **Response:** We appreciate the reviewer's insightful comments. The role of MMP-9 in the occurrence and development of fibrosis is indeed a subject of debate. This study (PMID: 35804363) has reported that MMP-2 and MMP-9 are not the main driving factors for pulmonary fibrosis. For confirmation, we examined leukocyte infiltration at the lesion site and the secretion of inflammatory factors in bleomycin-induced mouse lung tissues using IF and ELISA assay after administering Mexo-cl-aT via inhalation. As shown in **Fig. S18a**, the fluorescence intensity of CD45⁺ cells (a biomarker of leukocytes) was significantly higher in lung sections from the PBS group compared to a substantial decline observed in the Mexo-cl-aT group. In line with these observations, Mexo-cl-aT treatment resulted in a marked reduction in the

secretion of inflammatory cytokines, such as IL-6, TNF- α , and IL-1 β , demonstrating its robust capacity to inhibit leukocyte infiltration and the secretion of inflammatory cytokines in vivo. We have added the **Fig. S18** and corresponding description (line 305-307) in the revised manuscript.

Fig. S18 Anti-inflammation effect of Mexo-cl-aT. a. Immunofluorescence staining of CD45⁺ cells in mouse lung sections. Scale bar = 200 μ m. **b.** The levels of IL-6 (PBS vs. Mexo-cl-aT, $P < 0.0001$; $N=6$), TNF- α (PBS vs. Mexo-cl-aT, $P < 0.0001$; $N=6$), and IL-1 β (PBS vs. Mexo-cl-aT, $P = 0.2009$; $N=6$) in mouse lung tissues determined by ELISA. Data are means \pm SEM. ns, not significant; * $P < 0.05$; ** $P < 0.01$; **** $P < 0.0001$ (one-way ANOVA with Dunnett's multiple comparisons test).

- To further investigate the in vivo anti-fibrosis effects of Mexo-cl-aT, we added pirfenidone, a standard antifibrotic treatment, as a positive control and evaluated its therapeutic efficacy through HE and Masson staining. The healthy mice (control group) showed normal lung structure, whereas the bleomycin-induced pulmonary fibrosis mice (PBS group) exhibited severe lung damage. Both Mexo-cl-aT and pirfenidone treatments significantly ameliorated lung histopathology and reduced alveolar wall damage. Remarkably, the Mexo-cl-aT

group demonstrated superior antifibrotic effects compared to the pirfenidone group, as evidenced by a well-preserved alveolar structure and diminished collagen accumulation (**Fig. S17**). Correspondingly, Mexo-cl-aT treatment led to a marked decrease in the secretion of TGF- β 1 and TIMP-1, highlighting its robust anti-fibrotic capability. We have added the data as **Fig. S17** and corresponding description (line 305) in the revised manuscript.

Fig. S17 Anti-fibrosis effect of Mexo-cl-aT vs. pirfenidone in a bleomycin-induced pulmonary fibrosis mouse model. a. HE and masson staining of the lung sections of mice treated with different formulations. Scale bar = 200 μ m. **b.** The levels of TGF- β 1 (PBS vs. Mexo-cl-aT, $P = 0.0076$; $N=6$) and TIMP-1 (PBS vs. Mexo-cl-aT, $P <$

0.0001; N=6) in mouse lung tissues determined by ELISA. Data are means \pm SEM. ns, not significant; *P < 0.05; **P < 0.01; ***P < 0.001; ****P < 0.0001 (one-way ANOVA with Dunnett's multiple comparisons test).

2. Figure 1. The author correlated TIMP-1 with fibrosis in patients and fibrotic mice. However, I am much curious about the relation of MMP9 and Fibrosis. If there is an imbalance with TIMP-1 and MMP9, whether MMP9 would decrease within the fibrotic lung?

- **Response:** We thank the reviewer for drawing this point to our attention. Extensive research has demonstrated that the expression of MMP-9 is upregulated in pulmonary fibrosis (PMID: 29625182; PMID: 11983918; PMID: 10956632). Although the abundance of MMP-9 increases, with the increase of TIMP-1 expression, its activity is inhibited thereby it cannot fully exert the function of degrading collagen (PMID: 29518524; PMID: 32641389).
- To further assess the balance of ECM metabolism, BALF samples from healthy and fibrotic mice were collected, and the activities of TIMP-1 and MMP-9 were measured by ELISA. The results showed that TIMP-1 levels in mouse fibrotic lungs (PBS group) were significantly greater than those in healthy controls. Although free aT or Mexo reduced TIMP-1 levels and increased MMP-9 activity to some extent, this effect was enhanced when the two were combined (Mexo+aT, Mexo-fl-aT, and Mexo-cl-aT groups). As expected, TIMP-1 levels in the Mexo-cl-aT group decreased by 90.3% compared to those in the PBS group, approaching levels seen in the healthy control group (**Fig. 6g**). Accordingly, Mexo-cl-aT treatment resulted in the greatest increase in MMP-9 activity among all treatment groups (**Fig. 6h**). We have added the data as **Fig. 6g-h** and corresponding description (line 314-317) and experimental procedure (line 647-650) in the revised manuscript.

Fig.6g-h The level of TIMP-1 (g) and active MMP-9 (h) in mouse BALF (PBS vs. Mexo-cl-aT, $P < 0.0001$, $N=4$). Data are means \pm SEM; ns, not significant; * $P < 0.05$; ** $P < 0.01$; *** $P < 0.001$; **** $P < 0.0001$ (one-way ANOVA with Dunnett's multiple comparisons test).

3. Authors focused on fibroblasts and epithelial with in vitro experiments. While the in vivo experiments were much more complicated than the in vitro system. The author should provide more information about the cellular source of TIMP1 and MMP9, which would smooth the results to fibroblasts and epithelial. Besides, whether the anti-TIMP-1 treatment would increase the inflammatory cell infiltration into the lung?

- **Response:** Thanks for the reviewer's kind reminder. The cellular source of TIMP1 and MMP-9 has been extensively documented by other researchers. Fibroblasts are recognized as one of the cellular sources of MMP-9 (PMID: 36352225), while activated fibroblasts are identified as the primary source of TIMP-1 (PMID: 30243515). We have added the references as ref.12,25 in the revised manuscript.
- To investigate whether the Mexo-cl-aT treatment would increase the inflammatory response, we examined leukocyte infiltration at the lesion site and the secretion of inflammatory factors in bleomycin-induced mouse lung tissues using IF and ELISA assay after administering Mexo-cl-aT via inhalation. As shown in **Fig. S18a**, the fluorescence intensity of CD45⁺ cells (a biomarker of leukocytes) was significantly higher in lung sections from the PBS group compared to a substantial decline observed in the Mexo-cl-aT group. In line with

these observations, treatment with Mexo-cl-aT resulted in a marked reduction in the secretion of inflammatory cytokines, such as IL-6, TNF- α , and IL-1 β , demonstrating its robust capacity to inhibit leukocyte infiltration and the secretion of inflammatory cytokines *in vivo*. We have added the **Fig. S18** and corresponding description (line 305-307) in the revised manuscript.

Fig. S18 Anti-inflammation effect of Mexo-cl-aT. **a**. Immunofluorescence staining of CD45⁺ cells in mouse lung sections. Scale bar = 200 μ m. **b**. The levels of IL-6 (PBS vs. Mexo-cl-aT, $P < 0.0001$; $N=6$), TNF- α (PBS vs. Mexo-cl-aT, $P < 0.0001$; $N=6$), and IL-1 β (PBS vs. Mexo-cl-aT, $P = 0.2009$; $N=6$) in mouse lung tissues determined by ELISA. Data are means \pm SEM. ns, not significant; * $P < 0.05$; ** $P < 0.01$; **** $P < 0.0001$ (one-way ANOVA with Dunnett's multiple comparisons test).

4. Figure 6. I am quite curious about the effects when the nano system applied to the mice *in vivo*. Though the authors provide information about the anti-fibrotic effect, I will ask that what the effects that would be about the inflammation. Some related cytokines IL-1b, TNF-a, IL-6 and the most critical TGF-b expression should be detected to fulfill its anti-fibrotic effects.

- **Response:** Thanks for the reviewer's kind reminder. We assessed the levels of

inflammatory cytokines (e.g. IL-1 β , TNF- α , IL-6) and TGF- β in mouse lung tissues by ELISA. The results demonstrated that treatment with Mexo-cl-aT significantly reduced the secretion of inflammatory cytokines (**Fig. S18**). Furthermore, the expression of TGF- β in mouse lung tissues also showed a marked decrease (**Fig. S17**), indicating the robust anti-fibrotic capability of Mexo-cl-aT in vivo. We have added the data as **Fig. S17** and **Fig. S18** and

Fig. S17b. The levels of TGF- β 1 (PBS vs. Mexo-cl-aT, $P = 0.0076$; $N=6$) and TIMP-1 (PBS vs. Mexo-cl-aT, $P < 0.0001$; $N=6$) in mouse lung tissues determined by ELISA. Data are means \pm SEM. ns, not significant; * $P < 0.05$; ** $P < 0.01$; *** $P < 0.001$; **** $P < 0.0001$ (one-way ANOVA with Dunnett's multiple comparisons test).

Fig. S18b. The levels of IL-6 (PBS vs. Mexo-cl-aT, $P < 0.0001$; $N=6$), TNF- α (PBS vs. Mexo-cl-aT, $P < 0.0001$; $N=6$), and IL-1 β (PBS vs. Mexo-cl-aT, $P = 0.2009$; $N=6$) in mouse lung tissues determined by ELISA. $N=6$. Data are means \pm SEM. ns, not significant; * $P < 0.05$; ** $P < 0.01$; **** $P < 0.0001$ (one-way ANOVA with Dunnett's multiple comparisons test).

5. Figure 6. As for the Masson's and H&E staining, authors are expected to provide full lung lobe images with zoomed area for the readers better to evaluate the anti-fibrotic effects. The fibrosis in the lung were very heterogeneity. It would be better for the readers to evaluated the anti-fibrosis effects. Besides, IF staining to a-SMA and COL-1 were bad quality, confocal images would be better if the author could do that.

- **Response:** Thank the reviewer for the good suggestion. We have added full lung lobe images with zoomed area in **Fig. S14a** and **Fig. 6c**, and provided over-300 ppi IF images of a-SMA and COL-1 IF staining in **Fig. 6e** of the revised manuscript.

Fig. S14a H&E staining of mouse lungs with different treatments. Scale bar = 200 μm.

Fig. 6c. Masson staining of mouse lungs with different treatments. Scale bar = 200 μm.

Fig. 6e. IF staining of α -SMA and COL1A1 in mouse lung sections. Scale bar = 200 μ m.

6. Figure 6. The levels of TIMP1 and MMP9 in the BALF were not that convincing, since majority of the cytokines were in the lung interstitial. Also, there were huge variation when BAL was prepared. Western blot detecting the two important proteins with the lung tissue are needed to better illuminate the protein levels in the lung. The results would better illuminate the balance between the two proteins.

- **Response:** We thank the reviewer for the critical suggestions. To ensure the credibility of the data, we had recalibrated the protein concentrations of the samples by BCA assay before detecting the levels of TIMP-1 and MMP-9 in the BALF. The ELISA assay revealed a substantial reduction in TIMP-1 levels in the Mexo-cl-aT group, decreasing by 90.3% compared to the PBS group, and approaching the levels observed in the healthy control group (Fig. 6g). Correspondingly, Mexo-cl-aT treatment led to the most significant increase in MMP-9 activity across all treatment groups (Fig. 6h).

Fig. 6g-h. The level of TIMP-1 (g) and active MMP-9 (h) in mouse BALF (PBS vs. Mexo-cl-aT, $P < 0.0001$, $N=4$).

- **Response:** To elucidate the expression changes of various MMPs and provide a

more comprehensive understanding of the MMP-TIMP balance, we performed transcriptomic analysis. The results revealed a dominance of TIMPs in the PBS group. Treatment with Mexo-cl-aT significantly upregulated the expression of multiple MMPs, including MMP-1, MMP-2, MMP-3, MMP-7, MMP-8, MMP-9, and MMP-12. Moreover, the upregulation of MMPs surpassed that of TIMPs, indicating a shift towards extracellular matrix degradation. Additionally, there was a decrease in the expression of genes related to oxidative stress, suggesting effective alleviation of lung tissue damage. We have added the data as **Fig. S19** and corresponding description (line 317-320 and line 324-327) in the revised manuscript.

Fig. S19 Transcriptome analysis in the lung of mice with bleomycin-induced pulmonary fibrosis. a. Heatmap of MMP- and TIMP-related genes. **b.** Heatmap of oxidative stress-related genes. N=3. M-c-a: Mexo-cl-aT.

7. BLM induced pulmonary fibrosis would dissolve within 3-5 weeks after BML treated. If the author could provide more information about the Nano system against Crystalline silica induced fibrosis would be more attracting. Since CS particle would induced persist and progressive pulmonary fibrosis.

- **Response:** Thank the reviewers for their constructive comments. We further investigated the antifibrotic potential of Mexo-cl-aT in a mouse model of

assess the degree of pulmonary fibrosis (PMID: 18476815), based on HE and Masson staining. The results are presented in **Fig. 6d**. The ashcroft score of healthy lungs is within the range of 0-1. The score of bleomycin-challenged mice significantly increased, averaging 4.5. Different treatment groups showed varying degrees of alleviation of fibrosis. Among them, the Mexo-cl-aT group had the best effect, with a score of approximately 1.2.

Fig. 6d. Ashcroft score of mouse lungs after different treatments (PBS vs. Mexo-cl-aT, $P < 0.0001$; $N=6$). Data are means \pm SEM; ns, not significant; * $P < 0.05$; ** $P < 0.01$; *** $P < 0.001$; **** $P < 0.0001$ (one-way ANOVA with Dunnett's multiple comparisons test).

9. The anti-fibrotic effects would rely more on the anti-ROS system or the anti-TIMP-1 effects?

- **Response:** Thank the reviewers for their constructive comments. In fact, the synergistic interplay between anti-ROS and anti-TIMP-1 mechanisms underpins the efficacy of our anti-fibrosis therapy. Specifically, aT exhibits a targeted effect on restoring the functional capacity of MMPs for collagen degradation, while the anti-ROS system plays a fundamental role in improving the microenvironment for tissue repair. Our findings revealed a substantial reduction in TIMP-1 levels in the Mexo-cl-aT group, decreasing by 90.3% compared to the PBS group, and approaching the levels observed in the healthy control group (Fig. 6g). Correspondingly, Mexo-cl-aT treatment led to the most significant increase in MMP-9 activity across all treatment groups (Fig. 6h). Furthermore, flow cytometry analysis demonstrated elevated ROS levels in fibrotic lungs, which

were markedly reduced by an average of 76.8% following Mexo-cl-aT treatment (Fig. 6i).

Fig. 6 Anti-fibrotic effects of Mexo-cl-aT *in vivo*. **g-h.** The level of TIMP-1 (g) and active MMP-9 (h) in mouse BALF (PBS vs. Mexo-cl-aT, $P < 0.0001$, $N=4$). **i.** Flow cytometry analysis of the ROS levels in mouse lung tissues (PBS vs. Mexo-cl-aT, $P < 0.0001$, $N=5$). Data are means \pm SEM; ns, not significant; * $P < 0.05$; ** $P < 0.01$; *** $P < 0.001$; **** $P < 0.0001$ (one-way ANOVA with Dunnett's multiple comparisons test).

Reviewer #5 (Remarks to the Author):

Overall:

This manuscript describes a study deploying umbilical cord-derived mesenchymal exosomes decorated with ROS-cleavable antibodies to TIMP-1 and delivered intratracheally as a potential therapeutic nanosystem for idiopathic pulmonary fibrosis. While many aspects of the study are intriguing, the exact cellular and molecular targets of this complex nanosystem are poorly defined, and the data as presented do not clearly reveal therapeutic targets and mechanism of action. The use of a single model which does not recapitulate many aspects of the human disease limits the broad applicability of the findings.

- **Response:** We thank the reviewer for the positive comments and considering our work “**intriguing**”.

Major Points:

- The rationale for the focus on TIMP-1 and MMP9 specifically is poorly defined. Additional discussion is needed about which MMPs are implicated in IPF pathogenesis and why TIMP-1 was chosen (ie does it have higher affinity binding for these specific MMPs?)

- **Response:** Much thanks to the reviewers for the in-depth review of our manuscript. Previous studies and our study have shown that TIMP-1 is one of the key factors leading to collagen deposition in pulmonary fibrosis. Relative research (PMID: 28585723) demonstrated that TIMP-1 inhibit broad-spectrum MMPs (MMP-1, -2, -3, -7, -9, etc.), among which MMP-9 is most efficiently inhibited, K_i value < 50 pM. Therefore, our study focuses on the MMP-9/TIMP-1 rebalance. We have added the references as ref.13 in the revised manuscript.

- Figure 1 - TIMP staining – Figures are low resolution and lack use of additional cell markers (Krt5, aSMA etc) to determine location of TIMP expression. In human IPF images shown the staining is rather diffuse. Is it intracellular or extracellular? Their BAL ELISA would suggest extracellular, but their staining shows it in tissue - is it

being produced / secreted by specific cell types in the IPF lung? Can the authors use mouse lung tissue from their bleomycin experiment to look for TIMP levels via ELISA or Western blot?

- **Response:** Thanks for the reviewer's kind reminder. The cellular source of TIMP1 has been extensively documented by other researchers, and the activated fibroblasts are identified as the primary source of TIMP-1 (PMID: 30243515). We have added the references as ref.12 in the revised manuscript.
- Since TIMP-1 (<https://www.uniprot.org/uniprotkb/P12032/entry>) are secreted proteins that reside in the extracellular matrix, where aT can effectively exert its function. Specifically, aT neutralize free TIMP-1, leading to increased activity of MMP-9. For further confirmation, we examined performed IF staining for α -SMA and TIMP-1 in bleomycin-induced mouse lung tissues after administering Mexo-cl-aT via inhalation. As shown in **Fig. S1**, the fluorescence signals for α -SMA (red) did not colocalize with those for TIMP-1 (green), suggesting an extracellular localization of TIMP-1. Additionally, we assessed the levels of TIMP-1 in mouse lung tissues by ELISA. The results showed that TIMP-1 levels in the lung tissues of mice were approximately twice as high as those in the normal control group at day 10 post-bleomycin injury (**Fig. S2b**). We have added the **Fig. S1, Fig. S2b** and corresponding description (line 95-96 and line 102-105) in the revised manuscript.

Fig. S1 Immunofluorescence staining of α -SMA and TIMP-1 in human lung sections. Scale bar = 100 μ m.

Fig. S2b The level of TIMP-1 in mouse lung tissues ($P = 0.0154$, $N=3$). Control, healthy mouse; Bleomycin, bleomycin-induced pulmonary fibrosis mouse model. Data are means \pm SEM; ns, not significant; * $P < 0.05$ (one-way ANOVA with Dunnett's multiple comparisons test).

- Figure 1g/h – the method used to stain for “ROS” is not specified in the methods section. What was used? Very few reagents will react with all forms of ROS and are not stoichiometric. DCFH-DA was used later in the paper and may have been used here? If so, DCFH-DA is not a reliable measure of all ROS and has several limitations (e.g., PMID: 22027063). Additionally, staining fixed tissue sections for ROS is not ideal – reactive species are produced and reduced dynamically and fixation will not preserve these species. The authors should state these limitations clearly and use a complementary assay to assess redox dysregulation in these samples, such as the formation of oxidative byproducts or modifications (see PMID: 29988126). It is also unclear if human lung tissue was formalin fixed or frozen, as well as any demographic

information or source of IPF and healthy control lung tissues - these should also be specified in the methods.

- **Response:** Much thanks to the reviewers for the in-depth review of our manuscript. DCFH-DA is a widely used ROS probe in many studies (e.g. PMID: 32457361; PMID: 37933888). According to the reviewer's suggestion, we have a further understanding of the limitations of DCFH-DA in ROS detection. We repeated the relative experiments with a more reasonable ROS probe, Amplex Red. The results showed that Mexo-cl-aT significantly reduced H₂O₂ in the supernatant of bleomycin-treated MLFs. We have added the **Fig. S11**, experimental procedure in the supplementary information, and corresponding description (line 222-224) in the revised manuscript.

Fig. S11 Quantitative analysis of the H₂O₂ levels in the supernatant of bleomycin-treated MLFs (PBS vs. Mexo-cl-aT, P < 0.0001; N=3). Data are means ± SEM; ns, not significant; ****P < 0.0001 (one-way ANOVA with Dunnett's multiple comparisons test).

- Given that the reactive species are produced and reduced dynamically and lung tissue fixation may not preserve these species. we performed transcriptomic analysis to further assess redox dysregulation. The results revealed a dominance of oxidative stress in the PBS group. Treatment with Mexo-cl-aT significantly decreased the expression of multiple genes related to oxidative stress (e.g. GST, SOD, GPx), suggesting effective alleviation of lung tissue damage. We have added the data as **Fig. S19b** and corresponding description (line 325-327) in the revised manuscript.

- **Fig. S19 Transcriptome analysis in the lung of mice with bleomycin-induced pulmonary fibrosis. b.** Heatmap of oxidative stress-related genes. N=3. M-c-a: Mexo-cl-aT.

- Human lung tissue was formalin fixed. Demographic information of IPF and non-IPF lung tissues has been added as **Table S1** in the revised manuscript.

Table. S1 Demographic information of non-IPF and IPF lung tissues.

No.	Gender	Age	Diagnosis
Non-IPF			
1	Female	45	Atypical adenomatous hyperplasia
2	Male	70	Lung squamous cell carcinoma
3	Female	27	Lung adenocarcinoma
4	Female	42	Lung adenoma
5	Male	56	Benign pulmonary nodules
6	Male	77	Lung adenocarcinoma
7	Male	62	Benign pulmonary nodules
8	Female	57	Lung adenocarcinoma
9	Female	55	Benign pulmonary nodules
10	Female	53	Lung adenocarcinoma
IPF			
1	Male	51	Idiopathic pulmonary fibrosis
2	Female	73	Idiopathic pulmonary fibrosis
3	Male	54	Idiopathic pulmonary fibrosis
4	Male	62	Idiopathic pulmonary fibrosis
5	Female	54	Idiopathic pulmonary fibrosis
6	Male	48	Idiopathic pulmonary fibrosis
7	Male	56	Idiopathic pulmonary fibrosis
8	Female	60	Idiopathic pulmonary fibrosis

- Rationale for choice of mesenchymal exosomes (e.g., vs. an engineered nanoparticle) could be better developed. Is the goal to protect the antibody, or to only release it when ROS levels are high? Or is exosome delivery meant to reach intracellular space? Also, what is the rationale for sourcing exosomes from umbilical cord mesenchymal stem cells, and what additional molecular do these exosome preparations contain?

- **Response:** Thanks for the reviewer's kind reminder. MSC-derived exosomes (Mexo) are ideal nanocarriers for lung-targeted drug delivery via inhalation, given their optimal size (30–150 nm), biocompatibility, low immunogenicity, and inherent tropism for injured or inflamed tissues. Furthermore, they can transfer bioactive molecules (miRNAs, proteins, lipids) to recipient cells, exerting diverse therapeutic effects such as enhancing angiogenesis, reducing fibrosis, and stimulating cell proliferation and differentiation. Additionally, Mexo exhibit potent immunomodulatory properties by suppressing pro-inflammatory cytokines and promoting anti-inflammatory responses, emerging as a promising cell-free therapy for regenerative medicine, inflammation-related diseases, and organ injury (PMID: 32855385; PMID: 32939235). In our manuscript, we also demonstrate that Mexo serves not only as a delivery system but also as an active therapeutic ingredient, inhibiting fibroblast activation (Fig. 3a) and repairing the epithelium (Figs. 4d-g, S9) and endothelium (Fig. S10) in the treatment of pulmonary fibrosis. In the introduction of the revised manuscript (line 67-72), we have provided a comprehensive explanation for the selection of Mexo.
- **Response:** Exosomes from human umbilical cord mesenchymal stem cells (hUC-MSCs) are superior to those from other sources. First, they are extracted from the placenta, generally a discarded tissue, enabling an easy, non-invasive acquisition process with minimal ethical concerns (PMID: 26679929). Second, hUC-MSCs, originating from neonatal tissue, demonstrate a significantly faster proliferation rate than bone marrow mesenchymal stem cells (BM-MSCs) or adipose mesenchymal stem cells (AD-MSCs). The proliferation capacity of BM-MSCs diminishes with the age of the donor, while AD-MSCs are vulnerable

to influences such as obesity (PMID: 24428376). Finally, research has indicated that exosomes from hUC-MSCs possess enhanced therapeutic efficacy for lung injury treatment compared to exosomes from other sources (PMID: 39468662).

- Figure 3 – The authors treat MLF stimulated with TGF β with the exosome/TIMP antibody, but do not confirm that TIMP and/or MMP activity/levels were affected by TGF β treatment in these fibroblasts. Staining for TIMP/MMP or an activity assay in the TGF β treated fibroblasts would be helpful here. Also, in Fig3a, there is no information on number of fields of view assessed or replicates, or any quantification for these samples. It is hard to get an idea of the scale of the change from isolated images. These are cells in culture, so it should be possible to also perform western blotting and/or ELISAs of the targets investigated in this panel (aSMA, col1a1, and add TIMP)?

- **Response:** Thanks for the reviewer’s good suggestion. We determined the activity of MMP-9 and TIMP-1 in TGF- β 1-treated MLFs by ELISA. As shown in the **Fig. 4b-c**, the level of MMP-9 increased by two times, and the level of TIMP-1 increased by three times in TGF- β 1-activated MLFs compared to the control.

Fig. 4b-c The levels of TIMP-1 (PBS vs. Mexo-cl-aT, $P < 0.0001$; $N=3$) and active MMP-9 (PBS vs. Mexo-cl-aT, $P = 0.0077$; $N=3$) in MLF culture supernatants.

- In **Fig.3a**, we selected three or more fields of view for each sample.

Representative images are shown in **Fig.3a**. We also conducted quantitative experiments by flow cytometry (**Fig.3b-c**), which is more objective and persuasive than confocal microscopy.

Fig. 3 Anti-fibrotic effects of Mexo-cl-aT *in vitro*. **a.** IF staining of α-SMA and COL1A1 in MLFs. **b-c.** Flow cytometry analysis and quantification of α-SMA⁺ (PBS vs. Mexo-cl-aT, $P < 0.0001$) (b) and COL1A1⁺ cells (PBS vs. Mexo-cl-aT, $P < 0.0001$) (c) in MLFs (N=3). Data are means ± SEM; ns, not significant; * $P < 0.05$; *** $P < 0.001$; **** $P < 0.0001$ (one-way ANOVA with Dunnett's multiple comparisons test).

- Figure 3 – when MLF are treated with the exosome alone, there is significant reduction of aSMA and it looks like collagen (although it is not clear if this comparison was not significant or just not reported). What is the proposed explanation

for this? It looks like the exosome has equal effect to the antibody alone? In the scheme in Figure 1 it shows the proposed effect of Mexo alone acting on vascular endothelial cells and epithelial cells, but this point is not really discussed in the introduction or results.

- **Response:** Thanks for the reviewer’s kind reminder. Numerous studies, including those with PMID: 33845892 and PMID: 34344478, have documented that MSC-derived exosomes can inhibit fibroblast activation, thereby reducing α -SMA levels. Our findings align with these reports, as evidenced by the downregulation of α -SMA following Mexo treatment, as shown in **Fig. 3a-b**. Moreover, aT can neutralize free TIMP-1, restoring the functional capacity of MMPs for collagen degradation. Therefore, aT treatment also lead to the downregulation of α -SMA (**Fig. 3a-b**).

Fig. 3 Anti-fibrotic effects of Mexo-cl-aT *in vitro*. **a.** IF staining of α -SMA and COL1A1 in MLFs. **b-c.** Flow cytometry analysis and quantification of α -SMA⁺ (PBS vs. Mexo-cl-aT, $P < 0.0001$) (b) and COL1A1⁺ cells (PBS vs. Mexo-cl-aT, $P < 0.0001$) (c) in MLFs (N=3). Data are means \pm SEM; ns, not significant; * $P < 0.05$; *** $P < 0.001$; **** $P < 0.0001$ (one-way ANOVA with Dunnett's multiple comparisons test).

- Mesenchymal stem cell-derived exosomes (Mexo) serve as innate nano-carriers, adept at targeting inflammation/injury sites and augmenting drug retention at the lesion site while maintaining a superior biosafety profile. Furthermore, Mexo exhibits a spectrum of therapeutic properties, including anti-inflammatory and anti-fibrotic activities, as well as facilitation of tissue repair and regeneration. In the introduction of the revised manuscript (line 67-72), we have provided a comprehensive explanation for the selection of Mexo.

- Figure 4 in general – it seems like the Mexo-fl-aT and the Mexo-cl-aT both have the same effect on viability, apoptosis, and wound healing, with the only difference being that Mexo-fl-aT is not able to increase MMP9 activity. Would this indicate that these phenotypic effects on wound healing, apoptosis, cell viability, are not mediated through MMP9 activity? For some of the endpoints, like wound healing, it looks like the exosome alone increases the response.

- **Response:** We are grateful for the reviewer's insightful feedback. Indeed, Mexo is the primary mediator of anti-apoptotic and tissue repair effects. Consequently, both Mexo-fl-aT and Mexo-cl-aT exhibit identical impacts on cell viability, apoptosis, and wound healing. The primary function of MMP-9 activity is to facilitate collagen degradation, whereas its contribution to anti-apoptosis and tissue repair is relatively limited. Therefore, for some of the endpoints, it is the exosome play the main role in wound healing.

- Figure 4f - Again, DCFH-DA is not a reliable probe for ROS generation. Something like Amplex Red might be a better assay here, which also focuses on extracellular

H₂O₂ levels, and would better reflect the extracellular oxidative environment in which the Mexo-cl-aT would be cleaved.

- **Response:** According to the reviewer's suggestion, we repeated the relative experiments with a more reasonable assay, Amplex Red. The results showed that Mexo-cl-aT significantly reduced H₂O₂ in the supernatant of bleomycin-treated MLFs. We have added the **Fig. S11**, experimental procedure, and corresponding description (line 222-224) in the revised manuscript.

Fig. S11 Quantitative analysis of the H₂O₂ levels in the supernatant of bleomycin-treated MLFs. PBS vs. Mexo-cl-aT, $P < 0.0001$; $N=3$. Data are means \pm SEM; ns, not significant; **** $P < 0.0001$ (one-way ANOVA with Dunnett's multiple comparisons test).

- Figure 6g – TIMP1 levels decrease in response to these treatments, but even the Mexo alone condition causes this. What is the proposed mechanism for this? Does binding of TIMP-1 with the aT antibody cause it to be undetected by the ELISA? Also, this is in BAL- do tissue levels of TIMP-1 decrease as well? Can ELISA/staining/WB be performed on these?

- Can the authors demonstrate that when aT is released from Mexo-cl-aT, that the antibody then binds TIMP? Can this complex be identified and measured, ie via immunoprecipitation from in vivo samples? This would also help determine non-specific binding of aT to other targets (other TIMPs?) occurs in vivo.

- **Response:** We thank the reviewer for drawing this point to our attention. Several studies reported that exosomes derived from stem cells could inhibit fibroblast

activation (PMID: 33845892; PMID: 34344478), which may lead to decreased TIMP-1 secretion. Therefore, we can reasonably speculate that this explains why Mexo alone can reduce TIMP-1 levels.

- The molecular weight of TIMP-1 is about 23 kDa, and that of aT is about 150 kDa. When TIMP-1 is combined with aT, it has almost no excess binding site for another anti-TIMP-1 antibody, thereby undetected by the ELISA.
- We further quantified the levels of TIMP-1 in mouse lung tissues by ELISA. As shown in the **Fig. S17b**, the level of TIMP-1 in mouse lung tissues increased by 1.5 times following bleomycin inducement. Mexo-cl-aT treatment led to a significant decrease in TIMP-1 levels. We have added the **Fig. S17b** and corresponding description (line 304-305) in the revised manuscript.

Fig. S17b The levels of TIMP-1 in mouse lung tissues determined by ELISA. PBS vs. Mexo-cl-aT, $P < 0.0001$; $N=6$. Data are means \pm SEM. ns, not significant; ** $P < 0.01$; *** $P < 0.001$; **** $P < 0.0001$ (one-way ANOVA with Dunnett's multiple comparisons test).

- We appreciate your insightful question. However, due to the limitations of our current experimental setup, we are unable to directly demonstrate the binding of the antibody to TIMP upon the release of aT from Mexo-cl-aT. In the co-immunoprecipitation experiment, the magnetic beads coated with protein A/G non-discriminately captured both aT and mouse immunoglobulins, leading to impure co-precipitated samples and significant interference in the detection of the TIMP-1-aT complex.

- The discussion is not very detailed and the first two paragraphs merely summarize the results of the study, while only the last one discusses broader implications. What are the limitations of this study?

- **Response:** Thanks for the reviewer's kind reminder. Our research highlights the significant potential of Mexo-cl-aT as a novel treatment for IPF. We also recognize several challenges that remain for the clinical application of Mexo-cl-aT. These challenges encompass the limited large-scale production and high variability of exosomes—persistent issues in bio-optimization, industrialization, and standardization. Additionally, the antifibrotic potential of Mexo-cl-aT in a mouse model of crystalline silica (CS)-induced fibrosis was limited, with large, fused fibrotic nodules still persisting in the Mexo-cl-aT group (**Fig. S22**). This limitation is likely due to the difficulties in metabolizing CS particles in vivo, potentially leading to sustained and progressive pulmonary fibrosis. We have added the corresponding description (line 401-403) in the revised manuscript.

Fig. S22 Anti-fibrosis effect of Mexo-cl-aT in a silica-induced pulmonary fibrosis mouse model. a. Illustration of animal experiment procedure. **b.** Representative image, HE and masson staining of the lung sections of mice treated with different formulations. Arrows indicate fibrotic lesions. Scale bar = 200 μm .

- Overall, the main goal of the treatment is unclear – is the Mexo-cl-aT meant to deliver the anti-TIMP antibody and scavenge ROS through cleavage of the linker? Or is the Mexo component serving some value in wound repair? I do not think this distinction is well described in the introduction/results sections. In line 341 – 351 of the discussion this is mentioned, although the statement “Mexo have a natural ability to home to injury sites” is not cited. A more thorough introduction about why Mexo was chosen and what is known about exosomes ability to promote wound repair is needed in the introduction.

- **Response:** We are sorry for the confusion. In the revised manuscript (line 67-72), we have provided a comprehensive introduction explaining the rationale behind the selection of Mexo and the current understanding of exosomes' capacity to facilitate wound repair. The reasons for our choice of exosomes are outlined as follows: Firstly, exosomes serve as natural nano-carriers. Secondly, they possess the ability to target inflammation/injury sites and enhance the retention time of drugs at the lesion. Thirdly, MSC-derived exosomes exhibit therapeutic properties, including anti-inflammatory and anti-fibrotic activities. Lastly, exosomes offer superior biosafety due to their low immunogenicity.

Minor Points:

- Citations appear to be after the period for each sentence.
 - **Response:** We are sorry for this mistake. We replaced citations after the period for each sentence in the revised manuscript.
- Methods – “Determination of ROS in lung tissues” and “Determination of TIMP-1 in BALF” have the same description, which is for the TIMP measurement. ROS method section should be updated.
 - **Response:** We are sorry for this mistake. We have updated the methods (line 641-645) in the revised manuscript.
- Figure 2h – How was H₂O₂ quantified in these samples? This is not stated in the methods section.
 - **Response:** We are sorry for the neglect, and have added related content in the methods section of the revised manuscript. H₂O₂ was quantified by H₂O₂ assay kit according to the manufacturer's instructions. Briefly, H₂O₂ reacts with titanium sulfate to form a yellow titanium peroxide complex, which has characteristic absorption at 415 nm. Then, the absorbance at 415 nm of samples was measured using a microplate reader. We have added the methods (line 506-511) in the revised manuscript.

- The MMP9 activity assay in Figure 4B makes more sense to be included in Figure 3 – it is from the fibroblast experiment and involves the same treatments. The y-axis of this figure is confusing – is this activity or abundance (the ng/mL suggests abundance). This is also an issue in Figure 6C.

- **Response:** Thanks for the reviewer’s good suggestion. Our research methodology is outlined as follows: Figure 3 illustrates the overall anti-fibrotic effect of Mexo-cl-aT, whereas Figure 4 provides a detailed explanation of the functions of each component—aT activates MMP, Mexo facilitates tissue repair, and the phenylboronic acid ester bond is responsible for scavenging excess H₂O₂.
- The y-axis of Figure 6C (now Figure 6h) represents “activity of MMP-9”. This is because only active MMP-9 can be detected by ELISA, whereas MMP-9 inhibited by TIMP-1 remains undetectable. Accordingly, we revised the sentence “The level of MMP-9 in mouse BALF” to “The level of active MMP-9 in mouse BALF” in the revised manuscript (line 237, 349).

- Figure 4c – it is unclear what the treatment is. I’m assuming it is the bleomycin treatment, but no label is provided on the figure and no information on dose is given in the legend. The significance notation here is also unclear – is everything significant, or is it referring to a specific comparison?

- **Response:** We are sorry for this mistake. We have added the related information in **Fig.4f** legend in the revised manuscript. The significance notation here refers to a specific comparison (PBS vs. Mexo-cl-aT).

Fig. 4f Relative cell viability of bleomycin-challenged A549 cells after various

treatments for 24 ($P = 0.0014$), 48 ($P = 0.0001$), and 72 ($P = 0.0002$) hours ($N=3$).

- Figure 4d – I think having the graph shown in Figure S7a would be helpful here to compare the apoptotic cells in the same format as other assays

- **Response:** Thanks for the reviewer’s good suggestion. We have added the original **Fig. S7a** as new **Fig. 4e** to compare the apoptotic cells in the same format as other assays. we also added the corresponding description (line 205-206) in the revised manuscript.

- Mouse model is referred to as “IPF” – these mice don’t have IPF, they’ve been injured with bleomycin

- **Response:** We are sorry for this mistake. We have revised it to “a bleomycin-induced mouse model” in the revised manuscript.

Responses to the comments from the reviewers (Note: All the responses here are in blue. All the changes have been highlighted in red in the revised manuscript).

Reviewer #1 (Remarks to the Author):

Even though the author added RNA sequencing to evaluate the therapeutic effect of Mexo-cl-aT on pulmonary fibrosis, this only reflects changes at the genetic level, and the alterations in fibrosis marker proteins remain unknown. Additionally, could you provide the uncropped original membrane for Fig. 2A?

- **Response:** We are grateful for the reviewers' suggestions. As this study focused on TIMP-1, the balance of TIMP-1/MMP-9 was mainly evaluated. The other MMPs, TIMPs and redox-related proteins have not been explored in depth. We will improve this part in future studies. In Fig.6, the results showed that TIMP-1 levels in mouse fibrotic lungs (PBS group) were significantly greater than those in healthy controls. Although free aT or Mexo reduced TIMP-1 levels and increased MMP-9 activity to some extent, this effect was enhanced when the two were combined (Mexo+aT, Mexo-fl-aT, and Mexo-cl-aT groups). As expected, TIMP-1 levels in the Mexo-cl-aT group decreased by 90.3% compared to those in the PBS group, approaching levels seen in the healthy control group (Fig. 6g). Accordingly, Mexo-cl-aT treatment resulted in the greatest increase in MMP-9 activity among all treatment groups (Fig. 6h).

Fig. 6 Anti-fibrotic effects of Mexo-cl-aT *in vivo*. **a.** Illustration of animal experiment procedure. **b-d.** Representative images (**b**), Masson staining (**c**), and Ashcroft score (**d**) of mouse lungs after different treatments (PBS vs. Mexo-cl-aT, $P < 0.0001$; $N=6$). **e.** IF staining of α -SMA and COL1A1 in mouse lung sections. Scale bar = 200 μ m. **f.** Hydroxyproline content in mouse lung tissues (PBS vs. Mexo-cl-aT, $P < 0.0001$, $N=6$). **g-h.** The level of TIMP-1 (**g**) and active MMP-9 (**h**) in mouse BALF (PBS vs. Mexo-cl-aT, $P < 0.0001$, $N=4$). **i.** Flow cytometry analysis of the ROS levels in mouse lung tissues (PBS vs. Mexo-cl-aT, $P < 0.0001$, $N=5$). **j.** TUNEL staining of mouse lung sections. Data are means \pm SEM; ns, not significant; * $P < 0.05$; ** $P < 0.01$; *** $P < 0.001$; **** $P < 0.0001$.

0.0001 (one-way ANOVA with Dunnett's multiple comparisons test).

- The original membrane for Fig. 2a is as follows. The data was provided in a single Excel file.

Fig. 2a Original membrane of western blotting analysis.

Reviewer #3 (Remarks to the Author):

The authors conducted substantial experiments to prove their conclusions. However, I still have several major concerns.

- **Response:** We appreciate the reviewer's insightful comments. In response, we have conducted additional relevant experiments and implemented appropriate revisions to enhance the manuscript.

1. The first concern is about the cellular source of MMP9. Though there was report indicated that Fibroblasts are recognized as one of the cellular sources of MMP-9 (PMID:36352225), it was not in the fibrotic lung micro-environment. Still, the major cellular source of MMP9 is leucocytes, especially Macrophages. The author claimed that delivering Mexo-cl-aT would decrease CD45⁺ cells, as well as other pro-inflammatory cytokines. In that case, how the MMPs increased, what is the cellular source. We suppose that the MMPs came from fibroblast, the alleviated fibrosis did not support the hypothesis that it came from fibroblasts. The cellular source of increased MMPs should be explored in the circumstance. Extend experimental data is needed to support authors conclusion and for self-justification.

Overall, the decreased cytokine and CD45⁺ cells and fibroblast make me confused about how MMP9 increased in the lung microenvironment.

- **Response:** We sincerely thank the reviewer for raising this important point regarding the cellular source of MMP-9 and the apparent paradox between its elevation and reduced inflammation/fibrosis. We agree this requires clarification.
- Our data suggest a temporal sequence resolves this observation. In vitro: Active MMP-9 was detected early (4 hours post-Mexo-cl-aT treatment), while significant anti-fibrotic effects (quantification of α -SMA⁺ and COL1A1⁺ cells in MLFs) emerged later (24 hours). In vivo: Transcriptome analysis (MMP9/TIMP1) was performed at 24 hours post-intratracheal injection of Mexo-cl-aT, whereas key anti-fibrotic and anti-inflammatory endpoints (α -SMA/COL1A1 staining, Masson trichrome, cytokine levels, CD45⁺ cell counts) were assessed at 8 days.

- This staged timeline indicates that MMP-9 elevation is an early, transient event, potentially initiating ECM remodeling. In contrast, the sustained anti-fibrotic and anti-inflammatory effects arise from three complementary mechanisms: (1) the MMPs/TIMP-1 rebalancing by aT, (2) the potent tissue repair capabilities of Mexo, and (3) the ROS scavenging properties of the phenylboronic acid ester bond within Mexo-cl-aT. Therefore, the early MMP-9 increase is not contradictory to subsequent fibrosis and inflammatory alleviation; rather, it may be a prerequisite for beneficial ECM restructuring.
- We clarified this temporal framework in the Results section (lines 208-211) to emphasize that MMP-9 elevation represents an early, self-limiting event within a broader reparative cascade, directly addressing the reviewer's concern regarding its relationship with late-stage anti-fibrotic and anti-inflammatory outcomes.

2. The author believed the effects of the Mexo-cl-aT therapy depends on the anti-TIMP1 activity. However, the author did not intervene the TIMP1 in BLM-treated mice. In the Timp1 knockout mice, whether it could alleviate BLM-induced pulmonary fibrosis. If so, as reported in the previous reports, the most important issue is that whether Mexo-cl-aT treatment is still effective in the Timp1 knockout mice. This experiment can answer whether the treatment depends on the effect of TIMP1.

- **Response:** We thank the reviewers for their comments. Our key finding is that TIMP-1 knockdown significantly reduces fibrotic effects in mouse lung fibroblasts (MLFs). As demonstrated in **Figure S9**, transfection with different siRNA formulations targeting TIMP-1 resulted in a significant reduction in TIMP-1 expression, confirming successful gene silencing. Subsequently, immunofluorescence (IF) staining for the myofibroblast marker COL1A1 (**Figure S10**) revealed that TGF- β 1 treatment induced intense red fluorescence, indicating its pro-fibrotic effect. Critically, in TIMP-1-silenced MLFs (si-TIMP1), COL1A1 fluorescence was markedly reduced, irrespective of TGF- β 1 treatment. This result directly demonstrates that TIMP-1 knockdown attenuates fibrosis in this cellular model. We regret that we were unable to obtain TIMP-1 knockout mice for the in

vivo experiments. As an alternative approach, we focused on establishing the cellular mechanism using the robust siRNA knockdown model presented here. We have added the data as **Fig. S9-S10** and corresponding description (lines 205-208) in the revised manuscript.

Figure S9. Western blotting of TIMP-1 in MLFs after knockdown with different siRNAs.

Figure S10. Immunofluorescence staining of COL1A1 in control and TIMP-1-silenced MLFs, with or without TGF-β1 treatment. Scale bar = 100 μm.

3. Why is there was significant differences in the results between the positive control Pirfenidone and the treatment with Mexo-cl-AT? (FigS18) Why there were 8 groups in the elisa experiments, but 7 groups in the Immunofluorescence staining of CD45+ cells in mouse lung sections.

- **Response:** We are grateful for the reviewer’s attention to the experimental design details. These assays were specifically requested and added during the first revision as independent validation experiments. Notably, all essential comparison groups

required for the core analysis were consistently included in both the original and the supplementary experiments. Specifically, in **Figure S18** (now Figure S21), the additional group represents the Pirfenidone treatment group. We incorporated Pirfenidone, a standard antifibrotic treatment, as a positive control in this experiment to evaluate its therapeutic efficacy and further strengthen the validation of our findings.

4. ELISA analysis of TIMP-1 and MMP-9 would not evaluate the activity of TIMP1 and MMP9. It can only evaluate the protein contents. Gelatin enzyme spectrum experiment would demonstrate the enzyme activity.

- **Response:** Thank you for raising this important point regarding the assessment of TIMP-1 and MMP-9 activity. We appreciate your suggestion that gelatin zymography is the gold standard for directly evaluating MMP-9 enzymatic activity.
- Regarding our use of ELISA, we carefully consulted the manufacturer's instructions for the Mouse MMP-9 ELISA Kit (Elabscience, E-EL-M3052). These instructions specify that the assay is designed to detect the active form of MMP-9. MMP-9 molecules inhibited by TIMP-1 (or other inhibitors) or existing in the pro-form are not recognized by the antibodies in this kit. Therefore, while we acknowledge that ELISA primarily measures protein content, the specific design of this kit allows it to indirectly indicate active MMP-9 levels in the sample.
- We explicitly detailed the kit's brand and model (Elabscience, E-EL-M3052) in the Methods section of the revised manuscript. Additionally, we plan to incorporate gelatin zymography in future studies to complement these findings.

5. In FigS1, it seems that TIMP1 did not come from a-SMA+ fibroblasts. If the authors stained macrophages markers F4/80, it may be more co-localization.

- **Response:** We are very grateful to the reviewer for this exceptional insight and constructive suggestion. We have performed additional immunofluorescence co-staining experiments for TIMP1 and the macrophage marker CD68 on lung sections from IPF patients. It is worth noting that we selected CD68 instead of

F4/80 as the macrophage marker, since F4/80 is specifically expressed on murine macrophages and is not suitable for labeling macrophages in human IPF tissues.

- As shown in Figure S1b, the new results clearly show a substantial degree of colocalization of TIMP1 with CD68⁺ macrophages (highlighted by white arrows), indicating that macrophages are one of the cellular sources of TIMP1 in this context. At the same time, we also observed distinct TIMP1 signals located outside of CD68-positive cells, suggesting that other cell types, in addition to macrophages, may contribute to TIMP1 production during pulmonary fibrosis. These findings align with existing literature reports indicating that multiple cell types—with macrophages being particularly prominent—can express TIMP1 in IPF [1-3]. We apologize for not clearly stating this point in our original manuscript and thank the reviewer for prompting this clarification. The new image has been incorporated into the Figure. S1 in the revised supplementary materials.

Figure S1b. Immunofluorescence imaging of TIMP1 and CD68 expression in lung section from IPF patient.

Ref:

[1] Almutashiri, S; Alhumaid, A; Zhu, Y; et al. TIMP-1 and its potential diagnostic and prognostic value in pulmonary diseases. *Chinese Medical Journal Pulmonary and Critical Care Medicine*, 2023, 1, 67-76.

[2] Selman, M; Ruiz, V; Cabrera, S; et al. TIMP-1,-2,-3, and-4 in idiopathic pulmonary fibrosis. A prevailing nondegradative lung microenvironment? *American Journal of*

Physiology-Lung Cellular and Molecular Physiology, 2000, 279, 562-574.

[3] Kulshrestha, R; Pandey, A; Jaggi, A; et al. Beneficial effects of N-acetylcysteine on protease-antiprotease balance in attenuating bleomycin-induced pulmonary fibrosis in rats. *Iranian Journal of Basic Medical Sciences*, 2024, 366, 1-17.

6. The time points of Mexo-cl-aT intervention in vivo is still lack experimental evidence in Fig 6a. Confirmed in Fig S18, Mexo-cl-aT could exert anti-inflammatory efficacy during inflammatory stage. In addition, the ROS generation could also be observed in inflammatory stage. Please provide experiment data about Mexo-cl-aT intervention including inflammatory stage and fibrotic stage in IPF model, which could make the '10 day' time point rational.

- **Response:** We appreciate the reviewer's insightful query regarding the selection of Day 10 for Mexo-cl-aT intervention. Although fibrotic status of the lung tissues was significantly enhanced during 7-day bleomycin induction [1-3], our data demonstrate significantly higher collagen deposition and larger fibrotic areas at Day 10 versus Day 7, justifying this time point for modeling advanced disease (**Figure R1**).

Figure R1. Histological assessment of bleomycin-induced pulmonary fibrosis progression. Representative H&E-stained lung sections from mice at day 0, 7, and 10 post-intratracheal bleomycin instillation. Scale bar = 200 μ m.

- Our study specifically targets mid-to-late stage fibrosis—a critical unmet need in IPF treatment. The core innovation of Mexo-cl-aT lies in disrupting established fibrotic networks, and intervention at Day 10 directly addresses the therapeutic

challenge of reversing advanced fibrosis. This is evidenced by significant reduction in collagen deposition (Fig. 6c-d), downregulation of fibrotic (α -SMA, COL1A1; Fig. 6e), and restoration of MMP9/TIMP1 balance (Fig. 6g-h), indicating effective ECM remodeling. These findings demonstrate statistically significant improvements across fibrosis endpoints, validating the robustness of our experimental design.

Ref:

[1] Zhang, X., Dong, Y., Li, W. et al. Roxithromycin attenuates bleomycin-induced pulmonary fibrosis by targeting senescent cells. *Acta Pharmacol Sin.* 2021, 42, 2058-2068.

[2] Pan, L., Cheng, Y., Yang, W. et al. Nintedanib Ameliorates Bleomycin-Induced Pulmonary Fibrosis, Inflammation, Apoptosis, and Oxidative Stress by Modulating PI3K/Akt/mTOR Pathway in Mice. *Inflammation* 2023, 46, 1531-1542.

[3] Liu T, De Los Santos FG, Phan SH. The Bleomycin Model of Pulmonary Fibrosis. *Methods Mol Biol.* 2017, 1627, 27-42.

7. According to the Graphical Abstract, the core anti-fibrosis target of Mexo-cl-aT was regulating the balance between TIMP1 and active MMP9. However, the co-relation of Mexo-cl-aT and TIMP1 should be confirmed by TIMP1 encoding gene knockout mice, and the relationship between active MMP9 and TIMP1 also need demonstration. Following the Transcriptome analysis in Fig S19, Mexo-cl-aT treatment increased the expression of massive MMP family genes, which was contradictory against TIMP1-MMP9 balance. Thus, the 'Mexo-cl-aT-TIMP1-MMP9-anti-fibrosis' pathway could not be established with compelling evidence.

- **Response:** We sincerely thank the reviewer for this insightful critique regarding the establishment of the Mexo-cl-aT-TIMP-1-MMP-9-anti-fibrosis pathway. We acknowledge the limitation of not using TIMP-1 knockout mice in this study and deeply appreciate the suggestion to strengthen the causal relationship. To address this within our experimental constraints, we employed TIMP-1-silenced mouse

lung fibroblasts (MLFs) as an alternative model.

- Our data clearly demonstrate efficient TIMP-1 knockdown (**Figure S9**), which significantly reduced the expression of the key myofibroblast marker COL1A1 (**Figure S10**), directly linking TIMP-1 suppression to anti-fibrotic effects. Moreover, TIMP-1 silencing also resulted in a significant increase in active MMP-9 secreted by MLFs (**Figure S11**). These findings provide direct evidence supporting TIMP-1 as a key target for Mexo-cl-aT and its modulation of the TIMP-1/MMP-9 balance towards anti-fibrosis. We have added the data as **Fig. S9-S11** and corresponding description (lines 205-208) in the revised manuscript.

Figure S9. Western blotting of TIMP-1 in MLFs after knockdown with different siRNAs.

Figure S10. Immunofluorescence staining of COL1A1 in control and TIMP-1-silenced MLFs, with or without TGF-β1 treatment. Scale bar = 100 μm.

Fig. S11 Quantitative analysis of the active MMP-9 levels in cell lysates and supernatant of bleomycin-treated MLFs. PBS vs. TIMP-1-silenced MLFs; N=3. Data are means \pm SEM; ns, not significant; *P < 0.001 (one-way ANOVA with Dunnett's multiple comparisons test).**

- Regarding the transcriptome data (Fig. S19, now Fig. S22) showing upregulation of multiple MMP genes, previous studies (PMID: 28585723) have demonstrated that TIMP-1 inhibits a broad spectrum of MMPs (including MMP-1, -2, -3, -7, -9, etc.), with MMP-9 being the most efficiently inhibited ($K_i < 50$ pM). Therefore, our study focuses on the MMP-9/TIMP-1 rebalance. The transcriptome data revealing upregulation of multiple MMP genes after Mexo-cl-aT treatment does not contradict the TIMP-1/MMP-9 balance hypothesis. Instead, it highlights the complexity of MMP regulation: (1) TIMP-1 broadly inhibits MMPs, it is most potent against MMP-9. Thus, even if other MMPs are transcriptionally upregulated, their activity may remain suppressed by TIMP-1. (2) MMP activity is determined by the TIMPs/MMPs ratio, not MMPs expression alone. (3) The reduction in fibrosis markers (e.g., COL1A1 in TIMP-1-silenced MLFs) confirms that the net effect of TIMP-1 knockdown is anti-fibrotic, consistent with MMPs-mediated ECM degradation.

8. Although the authors weakened the content about MMP9 in the revised manuscript, the relationship between active MMP9 and reduced fibrosis in IPF should not be explained so simple, which was mentioned in my previous Q2. Please add some test index into in vivo experiment, which can better reflect upregulated collagen degradation specific governed by active MMP9.

- **Response:** We acknowledge the reviewer's valid concern regarding the complexity of active MMP9's role in collagen degradation during IPF fibrosis resolution.

While our transcriptomics data confirm broad upregulation of MMPs, we specifically focused on MMP9 in ELISA analyses because TIMP-1 inhibition most significantly enhanced MMP9 activity (as indicated by preliminary mechanistic studies, PMID: 28585723). This led us to propose MMP-9 as a potential key mediator in this context, though we fully recognize that the actual dynamics involve multifaceted interactions.

- To address your constructive suggestion for more specific indices of MMP-9-driven collagen degradation, we propose to incorporate the following approaches in future studies: (1) Direct MMP-9 activity assays (e.g., FRET-based probes, gelatin zymography) to quantify enzymatic function. (2) Collagen degradation fragment analysis (e.g., C1M for type I collagen, ICTP for crosslinked fragments). These measures would help establish causality between MMP-9 activity and fibrosis reduction, if feasible within our experimental framework. We are grateful for your guidance in strengthening the mechanistic rigor of our work and will carefully consider these additions in our revised discussion and future research plans.

9. In Fig S17, the pirfenidone treatment could not effectively hamper lung fibrosis in large field view, but TGF- β and TIMP1 reduced collectively. What's more, the level of IL-6 and TNF- α should be noticed in pirfenidone treated group in Fig S18.

- **Response:** We thank the reviewer for highlighting the importance of IL-6/TNF- α modulation in pirfenidone-treated groups. Our data show that pirfenidone reduced TGF- β /TIMP1 but failed to ameliorate fibrosis in large-field views (Fig S17, now Fig S20) or suppress IL-6/TNF- α (Fig S18b, now Fig S21b). This dissociation is consistent with pirfenidone's mechanism: it inhibits TGF- β -driven collagen synthesis but does not degrade mature ECM due to limited MMP activation (PMID: 35652284; PMID: 30611717), or block NF- κ B-mediated IL-6/TNF- α production (PMID: 35332068; PMID: 27498142). These limitations explain its partial clinical efficacy in IPF and underscore the advantage of Mexo-cl-aT. Our data demonstrate that Mexo-cl-aT surpasses pirfenidone by three complementary mechanisms: (1)

the MMPs/TIMP-1 rebalancing by aT, (2) the potent tissue repair capabilities of Mexo, and (3) the ROS scavenging properties of the phenylboronic acid ester bond within Mexo-cl-aT, thereby enabling superior suppression of inflammation, inhibition of TGF- β /TIMP1, and MMPs/TIMPs rebalance—leading to significant fibrosis resolution. The pirfenidone group thus serves as a valuable control highlighting the multi-targeted advantage of Mexo-cl-aT.

Reviewer #5 (Remarks to the Author):

Overall: The authors have responded to the reviewer concerns, and have provided extensive additional data and figures which have provided additional context and greatly improved the impact of the work.

- **Response:** We thank the reviewer for the positive comments.

Major Comments: none

Minor Comments:

1. Lines 67-72: Thank you for adding this helpful text. It is added in a paragraph discussing ROS, and it seems from the data that the phenylboronic ester bond, not the Mexo themselves, are what is scavenging ROS in this system. Please make the added text a separate paragraph.

- **Response:** Revised.

2. Fig. 4.g.: it would be helpful to add reference lines for “no migration” (PBS condition) to all the panels so that the comparable effects of treatments can be more easily seen. Also for Fig. S10c

- **Response:** Revised.

3. Line 307: “In a comparative study that pirfenidone was as the positive control”; suggest change to “In a comparative study in which pirfenidone was used as a positive control” to improve clarity.

- **Response:** Revised.

4. Line 403: “of exosomes, which are persistent issues in bio-optimization”; suggest change to “of exosomes-persistent issues in bio-optimization” for clarity.

- **Response:** Revised.

5. Line 404: “Additionally, the antifibrotic potential”; suggest change to “Furthermore, the approach may not be optimized for all forms of fibrosis. For example, the antifibrotic potential” for clarity.

- **Response:** Revised.